# DIMENSION-INDEPENDENT RATES FOR STRUCTURED NEURAL DENSITY ESTIMATION

## ABSTRACT

We show that deep neural networks achieve dimension-independent rates of convergence for learning structured densities such as those arising in image, audio, video, and text applications. More precisely, we show that neural networks with a simple $L^2$-minimizing loss achieve a rate of $n^{-1/(4+r)}$ in nonparametric density estimation when the underlying density is Markov to a graph whose maximum clique size is at most $r$, and we show that in the aforementioned applications, this size is typically constant, i.e., $r = O(1)$. We then show that the optimal rate in $L^1$ is $n^{-1/(2+r)}$ which, compared to the standard nonparametric rate of $n^{-1/(2+d)}$, shows that the effective dimension of such problems is the size of the largest clique in the Markov random field. These rates are independent of the data's ambient dimension, making them applicable to realistic models of image, sound, video, and text data. Our results provide a novel justification for deep learning's ability to circumvent the curse of dimensionality, demonstrating dimension-independent convergence rates in these contexts.

## 1 INTRODUCTION

Deep learning has emerged as a remarkably effective technique for numerous statistical problems that were historically extremely challenging, especially in high-dimensional settings. In the realm of deep generative models, which can be framed as density estimation, deep methods have showcased the ability to learn density functions with thousands or millions of dimensions using merely a few million data points (Oussidi & Elhassouny, 2018; Ho et al., 2020; Cao et al., 2024). This stands in stark contrast to standard density estimation theory, which would demand astronomical sample sizes due to the curse of dimensionality. The *manifold hypothesis* is perhaps the most widely accepted explanation for deep learning's ability to circumvent this curse (Bengio et al., 2013; Brahma et al., 2016). This hypothesis posits that, despite a distribution's ambient space being high-dimensional, the mass of the density is heavily concentrated around a lower-dimensional subset of that space, such as an embedded manifold. As we will argue later, for complex data types of significant interest—images, video, sound, and text—this assumption is intimately linked to spatio-temporal locality. For instance, covariates[1] that are nearby spatio-temporally, e.g., neighboring pixels, tend to be strongly dependent, suggesting they lie near a lower-dimensional subspace.

This paper investigates the benefits of leveraging the converse structure: The *independence* of spatio-temporally distant covariates. Covariates that are spatio-temporally distant often exhibit near-independence, particularly when conditioned on intervening covariates. Consider a sound recording: two one-second segments separated by a minute might share common elements, such as the same speaker. However, given the intervening minute of audio, these segments become effectively independent. The minute-long interval contains sufficient information to render the separated segments mutually uninformative. This principle of conditional independence extends to various data types, including images, where pixels far apart tend to be independent when conditioned on the surrounding region. This sort of dependence structure is naturally described by a *Markov random field* (MRF).

In this work, we show that, for a very general class of densities that are Markov to an undirected graph (a.k.a. MRF), density estimation can be achieved with a neural network and a simple $L^2$

---

[1] By "covariate" we mean a specific entry in a particular collection of data, for example a specific pixel location and color channel in a collection of image data or an index for tabular data.

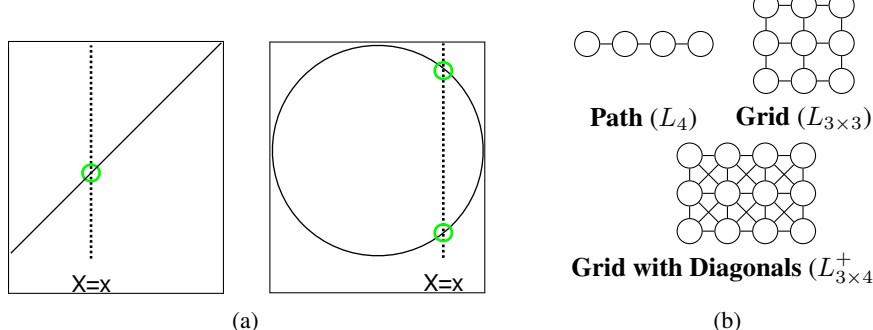

Figure 1: (a): Examples of conditioning $X = x$ (dotted lines) when a density's support is a manifold (solid lines). (b): *Highly simplified* examples of common MRF graphs. Paths correspond to sequential data and grids to spatial.

minimizing loss at a rate of approximately $n^{-1/(4+r)}$, where $r$ is the size of the largest clique in the graph. To compare, without the MRF assumption, the same class of densities can be estimated at a rate no better than $n^{-1/(2+d)}$, implying that the effective dimension is approximately $r$. We argue and show evidence that the MRF assumption is valid for many data types where neural networks excel (images, sound, video, etc.; see Figure 1b) and demonstrate that this approach to density estimation causes the effective dimension $r$ of these problems to remain constant or at least be orders of magnitude smaller than the ambient dimension $d$.

## 2 Background and Related Work

In this section, we lay the foundation for our main results by introducing key concepts and related work. We begin by discussing traditional approaches to nonparametric density estimation and their limitations, particularly the curse of dimensionality. We then explore the manifold hypothesis, a widely accepted explanation for the success of deep learning in high-dimensional settings. Following this, we introduce Markov random fields (MRFs) and their applications in modeling various types of data, including images and sequential information. This background will provide the necessary context for understanding the novelty of our approach, which leverages MRF structures to achieve dimension-independent convergence rates in density estimation, offering an alternative perspective to the manifold hypothesis.

### 2.1 Nonparametric Density Estimation

Density estimation is the task of estimating a $d$-dimensional target probability density $p$ from observed data, $\mathbf{x}_1, \ldots, \mathbf{x}_n \overset{\text{iid}}{\sim} p$. Of course, this is a classical problem for which we do not intend to provide a comprehensive overview, and instead refer readers to books such as Devroye & Gyorfi (1985); Devroye & Lugosi (2001); Tsybakov (2009) for additional background. Historically, $p$ was assumed to belong to a specific class of distributions, such as Gaussian, and the estimator $\hat{p}_n$ was selected accordingly. For more complex $p$, nonparametric density estimators like kernel density estimators or histograms are employed (Devroye & Gyorfi, 1985; Devroye & Lugosi, 2001). These methods converge to $p$ for *any* density given sufficient data, but notably suffer from the curse of dimensionality. For instance, when $p$ is Lipschitz continuous[2] and estimator parameters are optimally chosen, the $L^1$ error, $\int |p(x) - \hat{p}_n(x)|\,dx = \|p - \hat{p}_n\|_1$, converges at rate $O(n^{-1/(2+d)})$. This *nonparametric rate* is known to be optimal for Lipschitz continuous densities, and numerous studies over the past decade have established that neural networks and generative models can achieve this optimal rate (Liang, 2017; Singh et al., 2018; Uppal et al., 2019; Oko et al., 2023; Zhang et al., 2024; Kwon & Chae, 2024). This rate implies that the sample complexity grows *exponentially* in

---

[2]A function $f : \mathbb{R}^d \to \mathbb{R}$ is *Lipschitz continuous* if there exists $L \geq 0$ such that $|f(x) - f(y)| \leq L \|x - y\|_2$ for all $x, y$.

the dimension $d$, making the success of deep neural networks for estimating densities with millions of dimensions all the more remarkable. This is often explained via the *manifold hypothesis*.

## 2.2 MANIFOLD HYPOTHESIS

The manifold hypothesis posits that many high-dimensional real-world distributions concentrate around lower-dimensional spaces, such as submanifolds of the ambient space. This assumption underpins, either explicitly or implicitly, numerous machine learning methods. For instance, principal component analysis assumes that a distribution is concentrated around an affine subspace, while sparsity assumptions can be formulated as a union of manifolds, where the subsets of the union are axis-aligned subspaces.

The success of deep learning methods in handling high-dimensional data, such as images, videos, and audio, is frequently attributed to the manifold hypothesis. Experimental validation of the manifold hypothesis has been conducted for image datasets. In Pope et al. (2021), the authors determined that the intrinsic dimension of the ImageNet dataset lies between 25 and 40 dimensions, significantly lower than its ambient dimension. This hypothesis is closely linked to correlation and dependency between covariates. In images, for example, adjacent pixels $x_i$ and $x_j$ typically have similar values, causing the dataset to concentrate towards the linear subspace $x_i = x_j$, which is a submanifold. Figure 3(a) illustrates this concept, showing the values of pixels $(8, 8)$ and $(8, 9)$ for 100 randomly selected images from the grayscaled CIFAR-10 dataset, where a strong concentration along the diagonal is evident.

Further supporting this hypothesis, Carlsson et al. (2008) discovered that the set of $3 \times 3$ pixel patches from natural images concentrates around a 2-dimensional manifold. Theoretically, distributions concentrating around lower-dimensional subsets of the ambient space have been shown to yield improved estimation properties. For instance, Weed & Bach (2019) demonstrated that while a dataset typically converges at rate $n^{-1/d}$ to the true distribution in Wasserstein distance, when the dataset exhibits a lower $d'$-dimensional structure, it converges at the faster rate of $n^{-1/d'}$. Similar results illustrating the manifold hypothesis and its benefits can be found in Pelletier (2005); Ozakin & Gray (2009); Jiang (2017); Schmidt-Hieber (2019); Nakada & Imaizumi (2020); Berenfeld et al. (2022); Jiao et al. (2023); Tang & Yang (2024).

While the manifold hypothesis explains local dependencies, it's worth considering scenarios that deviate from this model. For example, the manifold hypothesis *cannot* be satisfied when covariates are independent. For example, if two covariates $x \sim p_x$ and $y \sim p_y$ are independent, their joint density $p_{x,y}(x, y) = p_x(x)p_y(y)$ fills a rectangle in their product space. As one may expect, pixels that are distant from one another tend to become more independent. This phenomenon is illustrated in Figure 3(d), which plots the grayscale values of pixels $(8, 8)$ and $(14, 28)$, showing a more dispersed pattern. This observation naturally leads to modeling the space of images as a Markov random field, where local dependencies are captured while allowing for independence between distant pixels.

**Remark 2.1.** A related result Cole & Lu (2024) shows that score-based diffusion can achieve dimension-free rates for estimating densities in the Barron space Barron (1993) with subgaussian tails.

## 2.3 MARKOV RANDOM FIELDS

A *Markov random field* (MRF) consists of a random vector $\mathbf{x} = (x_1, \ldots, x_d)$ and a graph $\mathcal{G} = (V, E)$, where the graph's vertices correspond to the entries of the random vector, i.e., $V = \{x_1, \ldots, x_d\}$. The graph encodes information about the conditional independence of the vector's entries. For a set $A = \{a_1, \ldots, a_{d'}\} \subset \{1, \ldots, d\}$, let $\mathbf{x}_A = (x_{a_1}, \ldots, x_{a_{d'}})$. Given three disjoint subsets $A, B, C$ of $\{x_1, \ldots, x_d\}$, the graph $\mathcal{G}$ indicates that the random vectors $\mathbf{x}_A$ and $\mathbf{x}_B$ are conditionally independent given $\mathbf{x}_C$ if there is no path from $A$ to $B$ that doesn't pass through $C$.

Consider a simple example with random variables x, y, and z, where $x = y + \epsilon_x$ and $z = y + \epsilon_z$, with $\epsilon_x, \epsilon_z$, and y being jointly independent. In this scenario, the distributions of x and z are conditionally independent given y. Figure 2 illustrates the corresponding MRF graph for this example.

It's important to understand that while an MRF conveys information about *conditional independence*, the absence of such information in the MRF does not necessarily imply dependence in the

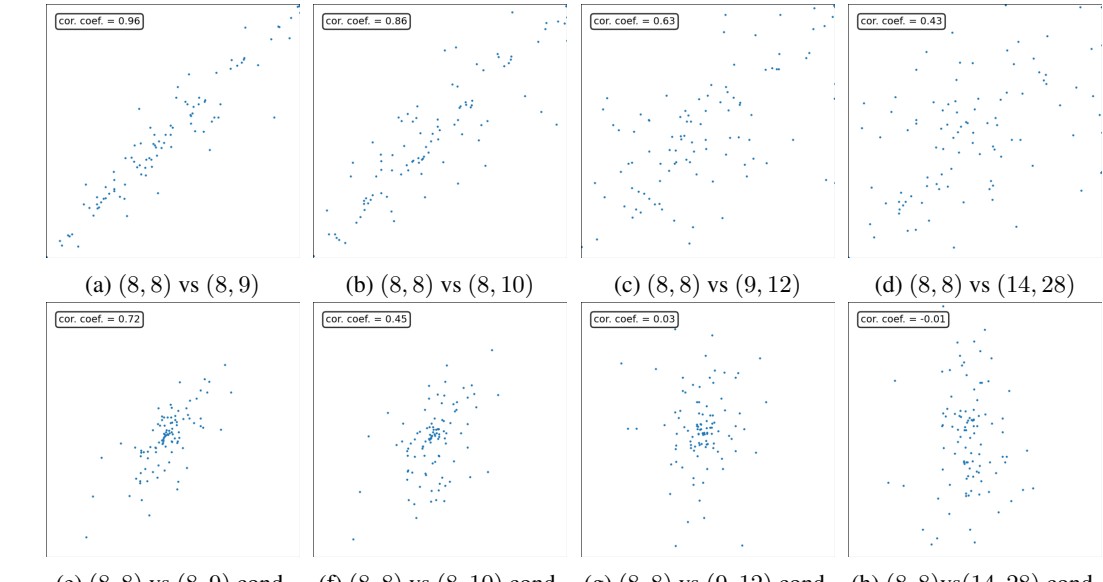

(a) $(8, 8)$ vs $(8, 9)$     (b) $(8, 8)$ vs $(8, 10)$     (c) $(8, 8)$ vs $(9, 12)$     (d) $(8, 8)$ vs $(14, 28)$

(e) $(8, 8)$ vs $(8, 9)$ cond.    (f) $(8, 8)$ vs $(8, 10)$ cond.    (g) $(8, 8)$ vs $(9, 12)$ cond.    (h) $(8, 8)$vs$(14, 28)$ cond.

Figure 3: Top row: Scatterplots comparing the grayscale values of pixel (8,8) with various other pixels for 100 randomly selected images. The decreasing correlation between pixels as their distance increases is evident.
Bottom row: The same comparisons as the top row, but conditioned on pixel (9,8) having a value approximately equal to 0.48 (the median value for this pixel across the dataset). Note the increased concentration of points towards the center along the horizontal axis, indicating reduced correlation when conditioned on a neighboring pixel.
These plots demonstrate how pixel correlations decrease with distance and how conditioning on a neighboring pixel can significantly reduce correlations, supporting the use of Markov Random Field models for image data. Similar plots for the COCO dataset can be found in Appendix F.

actual data. In other words, covariates can be conditionally independent in reality even if this independence is not explicitly represented in the MRF structure. The MRF provides a conservative model of independence relationships, capturing known or assumed conditional independencies without ruling out additional independencies that may exist in the data. Consequently, any random vector associated with a complete graph—where all vertices are adjacent to one another—is a valid MRF, since it provides no information about the independence of the covariates. This is because every vertex is connected to every other vertex, so removing any number of vertices will never separate the graph into multiple components. Because it conveys no information about the conditional independence of the covariates, it even applies to a random vector where all entries are independent.

One of the most well-known MRFs is the Markov chain. A Markov chain of length $d$ corresponds to the "path" graph $L_d$ of $d$ random variables. The above example with x, y, z corresponds to the graph $L_3$ and the MRF corresponding to $L_4$ shown in Figure 1b. In a Markov chain, the indices are often interpreted as a time parameter. A classic example is a gambling scenario: A person's money at time $t + 1$ is conditionally independent of their total value at time $t - s$ (for $s > 0$), given their value at time $t$. This property, known as the

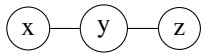

Figure 2: An example MRF. The random variables x and y are independent given z.

Markov property, encapsulates the idea that the future state depends only on the present state, not on past states. Markov chains have a long history of use for modeling sequential information, including audio and text data.

Beyond sequential data, MRFs have seen significant use in image processing. In this work, we focus on grayscale images for simplicity, bearing in mind that the results extend to RGB/color images as well. For image processing, the classic MRF model consists of a random variable that is a 2-dimensional grid $\mathbf{X} = [X_{i,j}]_{i,j}$ and a graph $\mathcal{G}$ where all pixels adjacent in $\mathbf{X}$ are also adjacent in

$\mathcal{G}$. Figure 1b contains one example from two different types of grid graphs: one standard "grid" graph $L_{3\times 3}$ and one "grid with diagonals" graph $L_{4\times 3}^{+}$, where the subscripts indicate the number of rows and columns of vertices, respectively. For the remainder of this work, our references to "grid" graphs encompass both variants—those with and without diagonal connections—unless explicitly stated otherwise. Such models have seen wide use in image processing and computer vision (see Li, 1994, for an overview). Denoising stands as perhaps the most common application of MRFs in image processing. This approach assumes that each pixel is best predicted using just its neighbors and ignoring the rest of the image. While this model proves effective for mitigating phenomena like additive white noise (Keener, 2010), it falls short as a comprehensive image model. Similarly, the path graph, often used for sequential data, oversimplifies the complex dependencies in real-world sequential information.

## 3 IMPROVING THE PATH AND GRID MARKOV RANDOM FIELD MODELS

While standard path and grid MRF models may suffice for correcting extremely local or high-frequency noise in sequential or spatial data, they fall far short of capturing the true distribution of complex data types. Consider, for example, audio data consisting of 21-second clips where the middle second is missing and needs to be predicted. According to the path MRF model, this missing second would depend solely on the audio samples directly preceding and following it. Consequently, under a Markov chain (i.e. path MRF) model, the remaining 20 seconds of audio (less two samples) would be deemed completely uninformative for predicting the middle second, given these two adjacent samples.

This simplistic model fails to capture the richer, longer-range dependencies present in real-world audio data. In practice, the content of the missing second is likely influenced by a broader context than just its immediate neighbors. For instance, the rhythm or theme established in the preceding few seconds, or the anticipation of what follows immediately after, could be crucial for predicting the missing segment. This moderately broader context is entirely discarded by the basic path MRF. Similarly, for image data, the standard grid MRF model suggests that a region of an image depends only on its immediate bordering pixels. However, realistic images often exhibit patterns and structures that span multiple pixels in various directions. For example, the edge of an object or a gradient in lighting might extend across several pixels, creating dependencies that the basic grid model fails to capture. Figure 4 illustrates this concept concretely, demonstrating the effects of different MRF models on image inpainting tasks and highlighting the implications of varying levels of contextual information. These limitations motivate the need for more sophisticated MRF models where segments or regions are more extensively connected, allowing for the incorporation of relevant contextual information without necessarily spanning the entire dataset.

To model sequential and spatial data more realistically, we propose using the "power graph" of the path and grid models. For a graph $\mathcal{G}$, the power graph $\mathcal{G}^t$ with $t \in \mathbb{N}$ is defined as the graph where an edge exists between every pair of vertices within $t$ steps of each other in $\mathcal{G}$, with $\mathcal{G}^1 = \mathcal{G}$. Figures 5 and 6 illustrate this concept using path graphs and grid graphs, respectively. This construction causes contiguous sections of sequences and patches of grids to become fully connected, as demonstrated in Figure 5.

Applying this power graph concept to a grid graph assumes that local patches of images are highly dependent, making no assumptions about conditional independence within a patch. It also implies that distant regions of an image become independent as the distance between them increases, and that these regions are independent when conditioned on a sufficiently wide separating region of pixels.

We can experimentally validate these assumptions using image data. The top row of Figure 3 shows the grayscale values of pixel (8,8) versus selected other pixels for 100 randomly chosen images from the CIFAR-10 training dataset. The bottom row repeats this experiment, but conditioned on the value of the adjacent pixel (9,8) being near its median value.

These experiments reveal that, when conditioned on the adjacent pixel, the dependence (as measured by correlation) decreases significantly. Notably, pixels (8,8) and (9,12) appear almost completely independent when conditioned on pixel (9,8). This provides strong evidence for the validity of the MRF model. The MRF model predicts that (8,8) and (9,12) should be independent when condi-

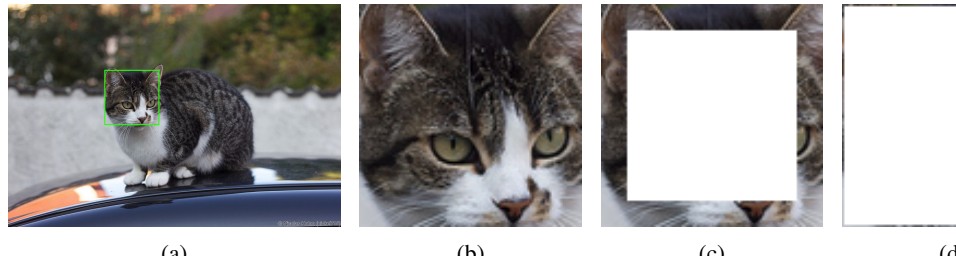

(a)  (b)  (c)  (d)

Figure 4: The leftmost image (a) is a $640 \times 427$ pixel photograph from the COCO 2014 dataset (Lin et al., 2014). Image (b) shows an enlarged version of the $102 \times 102$ pixel region outlined in (a). Images (c) and (d) display the 12-pixel and 1-pixel width borders of that region, respectively. Modeling this image with an MRF graph $L_{640 \times 427}$ or $L_{640 \times 427}^{+}$ would imply that the distribution of the missing interior in (d) depends exclusively on its 1-pixel wide border, with the rest of the image in (a) being uninformative for predicting this interior region. In contrast, predicting the interior using the 12-pixel border in (c) is more reasonable. This scenario corresponds to models like $L_{640 \times 427}^{6}$ or $\left(L_{640 \times 427}^{+}\right)^{6}$, which capture more extensive local dependencies. It's important to note that for the MRF model to hold, the interior doesn't need to be *deterministically* constructed from the surrounding pixels. Rather, the surrounding pixels need only provide sufficient information about the interior (e.g., that it's a cat's face) such that the rest of the image doesn't contribute any additional information for predicting the interior region.

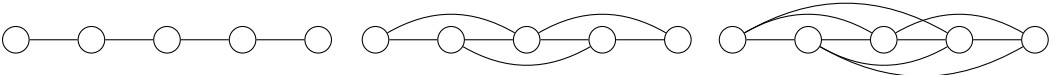

Figure 5: Illustrations of a path graph and its powers. Left: The path graph $L_5$. Center: The power graph $L_5^2$. Right: The power graph $L_5^3$. In $L_5$, only immediately contiguous vertices are connected. In $L_5^2$, every group of three contiguous vertices forms a complete subgraph. In $L_5^3$, every group of four contiguous vertices forms a complete subgraph. This progression demonstrates increasing connectivity among nearby vertices in the graph.

tioned on surrounding pixels (the number of which depends on the graph power of the MRF graph). Remarkably, we observe that pixels appear independent when conditioned on just a *single* adjacent pixel, suggesting that the grid MRF assumption may be even more conservative than necessary.

The power graph extension of path and grid MRFs presents a fundamentally different perspective on modeling high-dimensional data compared to the widely accepted manifold hypothesis. While the manifold hypothesis posits that high-dimensional data concentrates around lower-dimensional structures, our MRF approach embraces the full dimensionality of the data, focusing instead on the independence structure between variables. This model aligns well with the observed structure in various data types, capturing local dependencies while allowing for long-range independencies. For

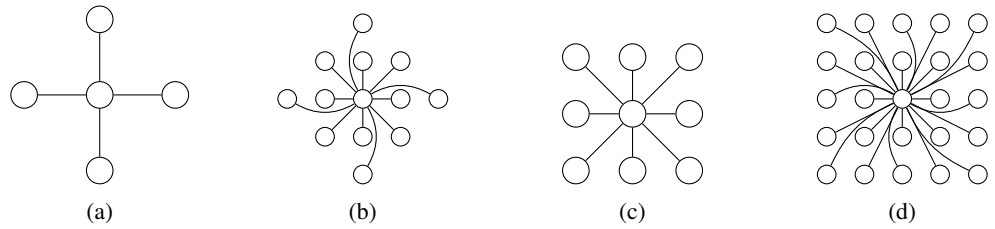

(a)  (b)  (c)  (d)

Figure 6: Comparison of vertex neighborhoods in different graph structures. (a) Neighborhood of a vertex in a standard grid graph $L_{d \times d}$. (b) Neighborhood of the same vertex in the power graph $L_{d \times d}^2$. (c) Neighborhood of a vertex in a grid graph with diagonals $L_{d \times d}^{+}$. (d) Neighborhood of the same vertex in the power graph $(L_{d \times d}^{+})^2$.

sequential data such as audio or text, it accounts for strong dependencies between nearby elements while acknowledging the decreasing influence of distant context. In spatial data like images, it models high correlation between neighboring pixels and gradual decorrelation as distance increases. Our experimental results provide compelling evidence for this MRF model's validity. The observed conditional independence between distant pixels, given intervening pixels, supports our power graph MRF approach's fundamental assumptions.

It's important to note that this approach is not meant to supersede the manifold hypothesis, but instead to augment it. The manifold hypothesis explains sample efficiency from local structure, while the MRF model adds additional model efficiency from a global perspective. Together, they provide a more comprehensive framework for understanding high-dimensional data.

Remarkably, in the following section, we will demonstrate that under these MRF assumptions, there exist estimators based on neural networks with standard loss functions (e.g. squared loss) that can achieve dimension-independent rates of convergence for density estimation. This result is particularly significant as it suggests a path to overcoming the curse of dimensionality in high-dimensional density estimation tasks without relying on low-dimensional embeddings. By focusing on independence structures rather than dimension reduction, our approach offers a novel explanation for the success of deep learning methods in processing complex, high-dimensional data, complementing and contrasting with the insights provided by the manifold hypothesis.

## 4 MRF-BASED DENSITY ESTIMATION WITH NEURAL NETWORKS

We begin by presenting the foundational results of this work that demonstrate that one can estimate a density $p$ given its Markov graph, at a rate that depends only on the size of the largest clique of the graph. We will present two results, one using a neural network style architecture using a practical empirical risk minimization style training and a second estimator that is more complex and computationally intractable that achieves approximately optimal rates of convergence.

### 4.1 STRUCTURED NEURAL DENSITY ESTIMATION

Our estimators are based on the classical Hammersley-Clifford Theorem (Hammersley & Clifford, 1971). Before presenting the theorem we must review a few concepts. A graph $\mathcal{G}$ is called *complete* if every vertex is adjacent to every other vertex. For a graph $\mathcal{G} = (V, E)$ a *clique* is a complete subgraph, i.e., $\mathcal{G}' = (V', E')$ with $V' \subset V$ and $E' \subset E$, that is complete. A *maximal clique* of a graph is a set of cliques which are not contained within another clique. Observe that maximal cliques of the same graph can have different numbers of vertices. See Figure 7 for examples of maximal cliques. The collection of the maximal cliques of a graph will be denoted $\mathcal{C}(\mathcal{G})$.

**Proposition 4.1** (Hammersley & Clifford, 1971). *Let $\mathcal{G} = (V, E)$ be a graph and $p$ be a probability density function satisfying the Markov property with respect to $\mathcal{G}$. Let $\mathcal{C}(\mathcal{G})$ be the set of maximal cliques in $\mathcal{G}$. Then*

$$p(\boldsymbol{x}) = \prod_{V' \in \mathcal{C}(\mathcal{G})} \psi_{V'}(\boldsymbol{x}_{V'}),$$

*where $\boldsymbol{x}_{V'}$ are the indices of $\boldsymbol{x}$ corresponding to $V'$.*

For neural networks we investigate estimators of the form:

$$\hat{p}(\boldsymbol{x}) = \prod_{V' \in \mathcal{C}(\mathcal{G})} \widehat{\psi}_{V'}(\boldsymbol{x}_{V'}),$$

where $\widehat{\psi}_{V'}$ are ReLU networks with architectures dependent only on $\mathcal{G}$, $|V'|$, and the sample size $n$. The weights are constrained to $[-1, 1]$, effectively implementing weight decay via constrained optimization rather than norm penalization.

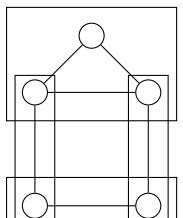

Figure 7: A graph with maximal cliques denoted by surrounding rectangles.

We analyze an estimator that minimizes the integrated squared error between $p$ and our estimator $\hat{p}$:

$$\int \left(p\left(\boldsymbol{x}\right) - \hat{p}\left(\boldsymbol{x}\right)\right)^2 d\boldsymbol{x} = \int p(\boldsymbol{x})^2 d\boldsymbol{x} - 2 \int p(\boldsymbol{x})\hat{p}(\boldsymbol{x})d\boldsymbol{x} + \int \hat{p}(\boldsymbol{x})^2 d\boldsymbol{x}. \tag{1}$$

In empirical minimization, the first term of equation 1 is constant. The second term can be estimated using the law of large numbers:

$$\int p(\boldsymbol{x})\hat{p}(\boldsymbol{x})d\boldsymbol{x} = \mathbb{E}_{\mathbf{x}\sim p}\left[\hat{p}(\mathbf{x})\right] \approx \frac{1}{n}\sum_{i=1}^{n}\hat{p}(\mathbf{x}_i) \quad \text{where } \mathbf{x}_1,\ldots,\mathbf{x}_n \overset{\text{i.i.d.}}{\sim} p.$$

The last term can be estimated stochastically. Let $U_d$ be the $d$-dimensional uniform distribution on the unit cube and $\epsilon_1, \epsilon_2, \ldots, \epsilon_{n'} \overset{\text{i.i.d.}}{\sim} U_d$. Then:

$$\int \hat{p}(\boldsymbol{x})^2 d\boldsymbol{x} = \mathbb{E}_{\epsilon\sim U_d}\left[\hat{p}(\epsilon)^2\right] \approx \frac{1}{n'}\sum_{i=1}^{n'}\hat{p}(\epsilon_i)^2.$$

### 4.2 MAIN RESULT

We now present our theorem on the convergence rate for $L^2$-minimizing neural network-based density estimators:

**Theorem 4.2.** *Let $\mathcal{G} = (V, E)$ be a finite graph and $r$ be the size of the largest clique in $\mathcal{G}$. There exists a known sequence of architectures $\mathcal{F}^*$ such that for*

$$\hat{p}_n = \arg\min_{f\in\mathcal{F}^*}\left(\|f\|_2^2 - \frac{2}{n}\sum_{i=1}^{n}f(\mathbf{x}_i)\right),$$

*where $\mathbf{x}_1,\ldots,\mathbf{x}_n \overset{\text{i.i.d.}}{\sim} p$, we have*

$$\|p - \hat{p}_n\|_1 \in \widetilde{O}_p\left(n^{-1/(4+r)}\right),$$

*for any Lipschitz continuous, positive density $p$ satisfying the Markov property with respect to $\mathcal{G}$.*

The proof of the theorem, based on results from Schmidt-Hieber (2017), details the architectures and specifies how their parameters scale with the sample size. The proof of this theorem, and all results in this work, can be found in the appendices. The minimax rate for density estimation on $d$-dimensional densities is $O\left(n^{-1/(2+d)}\right)$, so the "effective dimension" of an estimating a density using the estimator from the theorem above is $r + 2$. Consequently we see that the rate of convergence for density estimation can be greatly improved for MRFs with certain graphs $\mathcal{G}$. We will discuss the consequences of this in greater detail in Section 4.3.

### 4.3 CONSEQUENCES OF MAIN RESULTS

Our results indicate that the effective dimension of any density estimation problem under MRF assumptions is the size of its largest clique. The following results demonstrate examples where the largest clique is significantly smaller than the full dimensionality.

For definitions of the graphs $L_{d\times d'}$ and $L_{d\times d'}^+$, we refer the reader to the examples in Figure 1b. While these examples should provide intuitive understanding, formal definitions can be found in Appendix D.

**Images** We begin with the most compelling setting, corresponding to images:

**Lemma 4.3.** *Let $L_{d\times d'}$ be a $d \times d'$ grid graph with $t < d, d'$. The size of the largest clique in $L_{d\times d'}^t$ is less than or equal to $\frac{t^2+4t+3}{2}$.*

**Lemma 4.4.** *Let $L_{d\times d'}^+$ be the $d \times d'$ grid graph with diagonals, and $t < d, d'$. The size of the largest clique in the graph $\left(L_{d\times d'}^+\right)^t$ is $(t+1)^2$.*

Judging from the exmaple in Figure 3, $t = 2$ already gives a fairly reasonable model for images. Thus we have the following dimension-independent rate:

**Corollary 4.5** (Dimension-independent rates)**.** *The neural density estimator in Theorem 4.2 achieves a rate of*

$$\|p - \hat{p}_n\|_1 \in \widetilde{O}_p\left(n^{-1/7}\right) \quad \text{for the grid graph } L_{d \times d'}^2$$

*and*

$$\|p - \hat{p}_n\|_1 \in \widetilde{O}_p\left(n^{-1/9}\right) \quad \text{for the grid with diagonals graph } \left(L_{d \times d'}^+\right)^2.$$

Even when $t > 1$, we have $r = O(t^2)$ with $t \ll d$. In practice we expect $t = O(1)$, so even with $t > 1$, the rates are still dimension-independent.

Recall that if a density $p$ is an MRF with respect to a graph $\mathcal{G} = (V, E)$, it is also an MRF with respect to any graph $\mathcal{G}' = (V, E')$ that contains all the edges from $\mathcal{G}$, i.e., $E \subseteq E'$. Thus, the *absence* of edges in an MRF represents a stronger condition on $p$. In the graph $L_{d \times d'}^t$, every $(t+1) \times (t+1)$ block of vertices is fully connected. As demonstrated in Figure 3, when conditioned on an adjacent pixel, pixels tend to become independent with very little distance between them. Figure 3h shows that pixels (8,8) and (9,12) are seemingly independent conditioned on (9,8). Modeling CIFAR-10 as an MRF graph $L_{32 \times 32}^+$ would imply that (8,8) and (9,12) are independent conditioned on *every pixel surrounding (8,8)*, a much more stringent requirement than conditioning on one adjacent pixel. Thus, modeling CIFAR-10 as $(L_{32 \times 32}^+)^2$ appears to be a conservative approach. Consequently, the effective dimension for estimating CIFAR-10 is $(2+1)^2 = 9$ rather than $32 \times 32 = 1024$, an over 100-fold improvement!

**Sequences**    For sequential data, we have the following lemma:

**Lemma 4.6.** *Let $L_d$ be a $d$-length path graph. The size of the largest clique in $L_d^t$ is equal to $\min(t+1, d)$.*

Again, we observe that the effective dimension can be far less than the ambient dimension for sequential data, such as audio.

The MRF approach can be extended to various data types, yielding similar dimension reduction results. For instance, color images can be modeled as a three-dimensional random tensor $\mathbf{X} \in \mathbb{R}^{c \times w \times h}$ with a graph $\mathcal{G}$. In this model, the vertices in $\mathbf{X}_{:,i,j} \cup \mathbf{X}_{:,i',j'}$ are fully connected for $|i - i'| \le 1$ and $|j - j'| \le 1$, corresponding to a grid graph with diagonals where all channels are connected. Video data can be represented by four-dimensional graphs corresponding to order-4 tensors in $\mathbb{R}^{t \times c \times w \times h}$, with a similar connective structure. While text data is discrete in nature, once tokenized and passed through $d$-dimensional word embeddings, it resembles spatial data with dimensions $\mathbb{R}^{d \times t}$ and can benefit from independence structure.

In all these cases, the maximum clique size is determined by how quickly independence is achieved spatio-temporally or in the embedding space, rather than by the overall data dimensionality. This approach yields effective dimensions that are orders of magnitude smaller than the ambient dimension, leading to dimension-independent learning rates.

Crucially, this dimension independence is maintained across varying data sizes. For instance, cropping an image would leave the maximum clique size unchanged (provided the cropping isn't too extreme), while expanding an image would create a larger MRF graph but, assuming the underlying pattern holds, the maximum clique size would remain constant. This property results in a dimension-independent rate of learning that remains consistent across different image sizes. Thus, whether dealing with a $100 \times 100$ pixel image or a $1000 \times 1000$ pixel image of similar content, the effective learning rate remains tied to the maximum clique size rather than the total number of pixels, exemplifying true dimension independence in the learning process.

These extensions demonstrate the versatility of the MRF approach in modeling complex, high-dimensional data structures across various modalities, while significantly reducing the effective dimensionality of the problem.

**Hierarchical models**    Although not the primary focus of this work, our results have potential applications to other data types not typically associated with deep learning. For instance, hierarchical data is often modeled as a rooted tree. For tree-structured MRFs, the following is a well-known:

**Lemma 4.7.** *Let $\mathcal{G}$ be a tree with at least two vertices. The size of the largest clique in $\mathcal{G}$ is 2.*

Estimating densities with a tree MRF has been studied previously and is called "tree density estimation." The largest clique size being 2 and yielding a $\widetilde{O}(n^{-1/4})$ rate of convergence approximately matches previous work on this problem. In Liu et al. (2011); Györfi et al. (2022) it was found that one can estimate a density with an unknown tree MRF, without the strong density assumption at a rate $O(n^{-1/4})$. Compared to Theorem 4.2, this is an improvement by a factor of $n^2$, but these estimators are not based on neural networks, which is our focus. In the next section, we show that the $O(n^{-1/4})$ is not only optimal for trees, but can be generalized to arbitrary MRFs.

### 4.4 Approximately Optimal Estimator

Although not the primary focus of this work, we present the following result which approximately matches (up to $\log$ terms) the best possible rate for MRFs.

**Theorem 4.8.** *Let $\mathcal{G} = (V, E)$ be a finite graph. There exists an estimator $V_n$ such that for any Lipschitz continuous density $p$ satisfying the strong density assumption and Markov property with respect to $\mathcal{G}$, we have*

$$\|p - V_n\|_1 \in \widetilde{O}_p\left(n^{-1/(2+r)}\right),$$

*where $V_n$ is a function of $n$ i.i.d. samples from $p$, and $r$ is the size of the largest clique in $\mathcal{G}$.*

For this estimator, the effective dimension is $r$. The estimator analyzed in this theorem is based on Scheffé Tournaments over functions akin to histograms (Scheffe, 1947; Yatracos, 1985). This estimator is not computationally tractable and is presented solely to demonstrate the theoretical possibility of this optimal rate. An open question remains as to whether this optimal rate is achievable with neural networks and a tractable loss/algorithm. This rate cannot be substantially improved for any MRF graph which we argue in Appendix E.

## 5 Conclusion

Neural density estimation has been the subject of intense study over the past few decades, dating at least back to Magdon-Ismail & Atiya (1998). There has recently been interest in designing structured neural density estimators that exploit graphical structure (Germain et al., 2015; Johnson et al., 2016; Khemakhem et al., 2021; Wehenkel & Louppe, 2021; Chen et al., 2024). In this work, we have presented a novel perspective on the success of neural networks in density estimation problems. Our approach, based on Markov Random Field (MRF) structures, offers an alternative explanation to the widely accepted manifold hypothesis for why deep learning methods can circumvent the curse of dimensionality, and aligns with these recent developments on structured density estimation.

We have demonstrated that leveraging MRF assumptions can achieve dimension-independent convergence rates for density estimation, with the effective dimensionality determined by the largest clique in the MRF graph rather than the ambient data dimension. This potentially explains the efficacy of neural networks in domains like image and sequential data processing.

Our MRF-based approach complements, rather than replaces, the manifold hypothesis. We envision a combination of local manifold-like structures and global MRF-like independence properties at play in real-world scenarios, with the manifold hypothesis explaining local features and our MRF approach capturing broader independence structures.

This work opens avenues for future research, including investigating the interplay between local manifold structures and global MRF properties, and developing practical algorithms exploiting these structures within deep learning frameworks.

In conclusion, our work provides a novel theoretical framework for understanding neural networks in high-dimensional spaces, offering an alternative to the manifold hypothesis and potentially stimulating new directions in both the theory and practice of machine learning for high-dimensional tasks.

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

## A   NOTATIONS AND PRELIMINARIES

Before proving the main theorem we will first establish some notation and auxiliary results. For a pair of functions $f, g : \mathbb{X} \to \mathbb{R}$ where $\mathbb{X}$ is an arbitrary domain, we define the $f \cdot g$ to be pointwise function multiplication so $(f \cdot g)(x) = f(x)g(x)$ for all $x \in \mathbb{X}$. For a tuple of functions $f_1, \ldots, f_m : \mathbb{X} \to \mathbb{R}$, the product symbol $\prod_{i=1}^{m} f_i$ is defined to be pointwise function multiplication, i.e., $f_1(x) \cdot f_2(x) \cdot \cdots \cdot f_m(x)$ for all $x \in \mathbb{X}$. Let $\mathbb{N}$ be the set of positive integers. For any $d \in \mathbb{N}$, let $[d] = \{1, 2, \ldots, d\}$.

For a set $V \subset [d]$ with $V = \{v_1, \ldots, v_{|V|}\}$ where $v_i < v_j$ for all $i < j$, let $e_{d,V} : \mathbb{R}^d \to \mathbb{R}^{|V|}; x \mapsto [x_{v_1}, \ldots, x_{v_{|V|}}]$, i.e., $e_{d,V}$ accepts a $d$-dimensional vector and outputs the indices at $V$, in order. The function $e_{V,d}$ can be thought of as selecting some indices from a vector. As a slight abuse of notation, the $d$ subscript will be omitted.

For a graph $\mathcal{G} = (V, E)$, the set of maximal cliques in $\mathcal{G}$ will be denoted $\mathcal{C}(\mathcal{G})$, and is a set of subsets of $V$.

**All of our results will assume the domain of the data is the unit cube** $[0,1]^d$**.** A density $p$ will be called *positive* if $p(x) > 0$ for all $x \in [0,1]^d$. Since $[0,1]^d$ is compact, a direct consequence of this is that there exists $c > 0$ such that $p(x) > c$ for all $x$.

## A.1 PRELIMINARY RESULTS

**Proposition A.1.** *Let $p$ be a Lipschitz continuous probability density $[0,1]^d$, which is everywhere positive on $[0,1]^d$ and satisfies the Markov property with respect to a graph $\mathcal{G} = (V, E)$. Then, for all $x \in [0,1]^d$,*

$$p(x) = \prod_{V' \in \mathcal{C}(\mathcal{G})} \psi_{V'}\left(e_{V'}(x)\right),$$

*where each $\psi_{V'}$ is Lipschitz continuous, and there exist constants $c, C$ such that $0 < c \le C$ and $c \le \psi_{V'} \le C$ for all $V' \in \mathcal{C}(\mathcal{G})$.*

Before proving this proposition we first prove the following support lemma.

**Lemma A.2.** *Let $f, g : [0,1]^d \to \mathbb{R}$ be Lipschitz continuous with $f \ge \delta$ and $g \ge \delta$ for some $\delta > 0$. Then $f \cdot g$ and $1/f$ are both Lipschitz continuous and there exists $\delta' > 0$ such that $f \cdot g \ge \delta'$ and $1/f \ge \delta'$.*

*Proof of Lemma A.2.* Let $f$ be $L_f$-Lipschitz and $g$ be $L_g$-Lipschitz. Because $f$ and $g$ are Lipschitz on a bounded set there exists $C_f > 0$ and $C_g > 0$ such that $f \le C_f$ and $g \le C_g$. Let $x, y \in [0,1]^d$ be arbitrary.

We will begin by proving the product portion of the lemma:

$$
\begin{aligned}
|f(x)g(x) - f(y)g(y)| &\le |f(x)g(x) - f(x)g(y)| + |f(x)g(y) - f(y)g(y)| \\
&\le C_f |g(x) - g(y)| + C_g |f(x) - f(y)| \\
&\le C_f L_g \|x - y\|_2 + C_g L_f \|x - y\|_2 \\
&\le 2 \max\left(C_f L_g, C_g L_f\right) \|x - y\|_2 .
\end{aligned}
$$

The existence of a positive lower bound for $f \cdot g$ follows immediately from $f \cdot g \ge \delta^2$.

To prove the reciprocal portion of the lemma, observe that the function $x \mapsto 1/x$ is Lipschitz on the range of $f$. Since the composition of Lipschitz functions is itself Lipschitz, it follows that $1/f$ is Lipschitz. Finally we have that $1/f \ge 1/C_f > 0$, finishing the proof. $\quad\square$

*Proof of Proposition A.1.* This proof utilizes results from Chang (2007), a work-in-progress book currently used primarily as lecture notes. This work has a constructive proof of the Hammersley-Clifford Theorem. For $S \subset [d]$, let $\gamma_S : \mathbb{R}^d \to \mathbb{R}^d; x \mapsto [x_i \mathbf{1}(i \in S)]_i$, i.e., indices outside of $S$ are set to zero (the zero vector could actually be set to any arbitrary but fixed vector, e.g., set a vector $y \in \mathbb{R}^d$ and $[\gamma_S]_{i \notin S} = y_i$). From the Chang (2007) (3.15), the proof of the Hammersley-Clifford Theorem, it is shown that:

$$p(x) = \prod_{V' \in \mathcal{C}(\mathcal{G})} \psi_{V'}(x),$$

where,

$$\psi_{V'}(x) = \frac{\prod_{V'' \subset V' : |V'' \setminus V'| \mod 2 = 0} p\left(\gamma_{V''}(x)\right)}{\prod_{V'' \subset V' : |V'' \setminus V'| \mod 2 = 1} p\left(\gamma_{V''}(x)\right)}. \tag{2}$$

For clarity we do an example of the index set of the product; so

$$V'' \subset V' : |V'' \setminus V'| \mod 2 = 0$$

denotes a product over all subsets of $V'$ where the set $V''$ where $V'' \cap V'^C$ contains an even number of elements. From equation 2 it is clear that $\psi_{V'}$ only depends on the indices of $x$ in $V'$. The regularity conditions hold due to Lemma A.2.

$\square$

Given any set $S$. For any subset $S' \subseteq S$, let $\mathbf{1}_{S'}$ be the indicator function from $S$ to $\{0, 1\}$, i.e.

$$\mathbf{1}_{S'}(x) = \begin{cases} 0 & \text{if } x \notin S' \\ 1 & \text{if } x \in S' \end{cases} \quad \text{for all } x \in S.$$

Given any $d, b \in \mathbb{N}$, $V = \{v_1, \ldots, v_{|V|}\} \subset [d]$ and $C \geq 1$. For any $A \in [b]^{|V|}$, let $\Lambda_{d,b,A,V}$ be the subset of $[0, 1]^d$

$$\Lambda_{d,b,A,V} := \left\{ x \in [0, 1]^d \mid x_{v_i} \in \left[ \frac{A_i - 1}{b}, \frac{A_i}{b} \right] \quad \text{for all } i \in [|V|] \right\}. \tag{3}$$

Let $\mathbb{Q}_{d,b,V,C}$ be the set of functions from $[0, 1]^d \rightarrow \mathbb{R}$

$$\mathbb{Q}_{d,b,V,C} := \left\{ x \mapsto \sum_{A \in [b]^{|V|}} w_A \mathbf{1}_{\Lambda_{d,b,A,V}}(x) \mid w_A \in [0, C] \right\}. \tag{4}$$

For a set $\mathcal{L} \subset L^\beta \left( [0, 1]^d \right)$ where $1 \leq \beta < \infty$ and $\epsilon > 0$, a subset $\mathcal{C} \subseteq \mathcal{L}$ is called an $\epsilon$-cover of $\mathcal{L}$ in $L^\beta$ norm if, for any $f \in \mathcal{L}$, there exists a $g \in \mathcal{C}$ such that $\|f - g\|_\beta \leq \epsilon$. Also, we define $N(\mathcal{L}, \epsilon)$ to be the cardinality of the smallest subset of $\mathcal{L}$ that is a (closed) $\epsilon$-cover of $\mathcal{L}$ in $L^\beta$ norm. Note that $N(\mathcal{L}, \epsilon)$ depends on $\beta$. We will not specify it when it is clear in the context.

# B    PROOF OF THEOREM 4.2

**Theorem B.1.** *Let $\mathcal{G} = (V, E)$ be a finite graph and $r$ is the size of the largest clique in $\mathcal{G}$. There exists a neural network architecture $\mathcal{F}^*$, such that, for*

$$\hat{p}_n = \arg \min_{f \in \mathcal{F}^*} \|f\|_2^2 - \frac{2}{n} \sum_{i=1}^n f(X_i)$$

*where $X_1, \ldots, X_n \overset{i.i.d.}{\sim} p$, then*

$$\|p - \hat{p}_n\|_2 \in \widetilde{O}_p \left( n^{-1/(4+r)} \right),$$

*for any Lipschitz continuous, positive density $p$ satisfying the Markov property with $\mathcal{G}$.*

This is stronger than $L^1$ convergence since, through Hölder's inequality, we get $L^1$ convergence at the same rate.

**Lemma B.2.** *Let $(\Omega, \Sigma, \mu)$ be a measure space, and let $f_1, \ldots, f_m$ and $g_1, \ldots, g_m$ be measurable and absolutely integrable functions on $\Omega$. Further suppose there exists a constant $C \geq 0$ such that, for all $i \in [m]$,*

$$\|f_i\|_\infty \leq C \quad and \quad \|g_i\|_\infty \leq C.$$

*Then the following inequality holds:*

$$\left\| \prod_{i=1}^m f_i - \prod_{i=1}^m g_i \right\|_\infty \leq C^{m-1} \sum_{i=1}^m \|f_i - g_i\|_\infty.$$

*Proof of Lemma B.2.* We will proceed by induction on $m$.

**Case** $m = 1$: Trivial.

**Induction:** Suppose the lemma holds for some value of $m$. From the inductive hypothesis we have

that

$$\left\|\prod_{i=1}^{m+1} f_i - \prod_{i=1}^{m+1} g_i\right\|_\infty \leq \left\|\prod_{i=1}^{m} f_i \cdot f_{m+1} - \prod_{i=1}^{m} g_i \cdot f_{m+1}\right\|_\infty + \left\|\prod_{i=1}^{m} g_i \cdot f_{m+1} - \prod_{i=1}^{m} g_i \cdot g_{m+1}\right\|_\infty$$

$$\leq \left\|\prod_{i=1}^{m} f_i - \prod_{i=1}^{m} g_i\right\|_\infty \|f_{m+1}\|_\infty + \left\|\prod_{i=1}^{m} g_i\right\|_\infty \|f_{m+1} - g_{m+1}\|_\infty$$

$$\leq \left\|\prod_{i=1}^{m} f_i - \prod_{i=1}^{m} g_i\right\|_\infty C + C^m \|f_{m+1} - g_{m+1}\|_\infty$$

$$\leq C^{m-1} \sum_{i=1}^{m} \|f_i - g_i\|_\infty C + C^m \|f_{m+1} - g_{m+1}\|_\infty$$

$$\leq C^m \sum_{i=1}^{m+1} \|f_i - g_i\|_\infty .$$

$\square$

**Space of Neural Network Architectures**   Define a space of neural networks as follows. Let $\sigma$ be the ReLU activation function with will act element-wise on vectors. For any $\ell \in \mathbb{N}$, $w = (w_0, \ldots, w_{\ell+1})$ with $w_i \in \mathbb{N}$, $s \in \mathbb{N}$ and $F > 0$, the space $\mathcal{F}(\ell, w, s, F)$ is defined by the functions $f : [0,1]^{w_0} \to \mathbb{R}^{w_{\ell+1}}$ which have the form:

$$f(x) = W_\ell \sigma_{v_\ell} W_{\ell-1} \sigma_{v_{\ell-1}} \cdots W_1 \sigma_{v_1} W_0 x,$$

where $\sigma_{v_i}(y) = \sigma(y - v_i)$, $W_i \in \mathbb{R}^{w_{i+1} \times w_i}$, where every entry in $W_i$ and $v_i$ have absolute value less than or equal to 1, $\|f\|_\infty \leq F$, and sum of the total number of nonzero entries of $W_i$ and $v_i$ is less than or equal to $s$. In this work the output dimension of all neural networks will be 1, i.e. $w_{\ell+1}$ will always be assumed to be 1. This is the same space of neural network models employed by Schmidt-Hieber (2017).

**Theorem B.3** (Theorem 5, Schmidt-Hieber, 2017). *For any $f \in C_d^\beta([0,1]^d, K)$ and any integers $m \geq 1$ and $N \geq \max((\beta+1)^d, (K+1)e^d)$, there exists a ReLU network $\tilde{f} \in \mathcal{F}(\ell, w, s, \infty)$ with depth*

$$\ell = 8 + (m+5)(1 + \lceil \log_2(\max(d, \beta)) \rceil), \tag{5}$$

*widths*

$$w = (d, 6(d + \lceil \beta \rceil)N, \ldots, 6(d + \lceil \beta \rceil)N, 1), \tag{6}$$

*and sparsity*

$$s \leq 141(d + \beta + 1)^{d+3} N(m + 6) \tag{7}$$

*such that*

$$\left\|\tilde{f} - f\right\|_{L^\infty([0,1]^d)} \leq (2K+1)(1 + d^2 + \beta^2)6^d N 2^{-m} + K 3^\beta N^{-\beta/d}. \tag{8}$$

**Lemma B.4** (Lemma 5, Remark 1, Schmidt-Hieber, 2017). *For any $\delta > 0$,*

$$\log N(\mathcal{F}(\ell, w, s, \infty), \epsilon, \|\cdot\|_\infty) \leq (s+1) \log(2^{2\ell+5} \epsilon^{-1}(\ell+1) w_0^2 w_{\ell+1}^2 s^{2\ell}).$$

### B.1   SQUARED-$L^2$ ERSION

*Proof of Theorem B.1.*   Recall that, given a graph $\mathcal{G}$, $\mathcal{C}(\mathcal{G})$ is the set of maximal cliques in $\mathcal{G}$. For any $V' \in \mathcal{C}(\mathcal{G})$, let $\mathcal{F}_{V'} = \mathcal{F}(\ell_{V'}, w_{V'}, s, C)$ where $\ell_{V'}, w_{V'}, s, C$ will be determined later. Also, let

$$\mathcal{F}^* = \left\{ \prod_{V' \in \mathcal{C}(\mathcal{G})} q_{V'} \circ e_{V'} \,\Big|\, q_{V'} \in \mathcal{F}_{V'} \right\}. \tag{9}$$

We shall show that $\mathcal{F}^*$ is the neural network architecture satisfying the desired guarantees in Theorem B.1.

For any set of $n$ i.i.d. samples $X_1, \ldots, X_n$ drawn from $p$, let

$$p_n^* = \arg\min_{f \in \mathcal{F}^*} \|p - f\|_2^2 \quad \text{and} \quad \hat{p}_n = \arg\min_{f \in \mathcal{F}^*} \left( \|f\|_2^2 - \frac{2}{n} \sum_{i=1}^n f(X_i) \right). \tag{10}$$

Now, we would like to bound the term $\|\hat{p}_n - p\|_2^2$. We first express it as

$$\|\hat{p}_n - p\|_2^2 = \left( \|\hat{p}_n - p\|_2^2 - \|p_n^* - p\|_2^2 \right) + \|p_n^* - p\|_2^2.$$

For the term $\|\hat{p}_n - p\|_2^2 - \|p_n^* - p\|_2^2$, we further express it as

$$\|\hat{p}_n - p\|_2^2 - \|p_n^* - p\|_2^2 = \underbrace{\|\hat{p}_n - p\|_2^2 - \left( \|p\|_2^2 + \|\hat{p}_n\|_2^2 - \frac{2}{n} \sum_{i=1}^n \hat{p}_n(X_i) \right)}_{:=A}$$

$$+ \underbrace{\left( \|p\|_2^2 + \|\hat{p}_n\|_2^2 - \frac{2}{n} \sum_{i=1}^n \hat{p}_n(X_i) \right) - \|p_n^* - p\|_2^2}_{:=B} \tag{11}$$

Before we bound $A$ and $B$, we first provide a useful inequality. For any $p' \in \mathcal{F}^*$, we have

$$\|p' - p\|_2^2 - \left( \|p\|_2^2 + \|p'\|_2^2 - \frac{2}{n} \sum_{i=1}^n p'(X_i) \right)$$

$$= \frac{2}{n} \sum_{i=1}^n p'(X_i) - 2 \langle p', p \rangle$$

$$\leq 2 \max_{f \in \mathcal{F}^*} \left| \mathbb{E}_p(f) - \frac{1}{n} \sum_{i=1}^n f(X_i) \right| \quad \text{since } \langle p', p \rangle = \mathbb{E}_p(p') \text{ and } p' \in \mathcal{F}^*. \tag{12}$$

For the term $A$, we immediately have

$$A = \|\hat{p}_n - p\|_2^2 - \left( \|p\|_2^2 + \|\hat{p}_n\|_2^2 - \frac{2}{n} \sum_{i=1}^n \hat{p}_n(X_i) \right)$$

$$\leq 2 \max_{f \in \mathcal{F}^*} \left| \mathbb{E}_p(f) - \frac{1}{n} \sum_{i=1}^n f(X_i) \right| \quad \text{since } \hat{p}_n \in \mathcal{F}^*.$$

For the term $B$, we have

$$B = \left( \|p\|_2^2 + \|\hat{p}_n\|_2^2 - \frac{2}{n} \sum_{i=1}^n \hat{p}_n(X_i) \right) - \|p_n^* - p\|_2^2$$

$$\leq \left( \|p\|_2^2 + \|p_n^*\|_2^2 - \frac{2}{n} \sum_{i=1}^n p_n^*(X_i) \right) - \|p_n^* - p\|_2^2 \quad \text{by the optimality of } \hat{p}_n$$

$$\leq 2 \max_{f \in \mathcal{F}^*} \left| \mathbb{E}_p(f) - \frac{1}{n} \sum_{i=1}^n f(X_i) \right| \quad \text{since } \hat{p}_n \in \mathcal{F}^*.$$

By plugging them into equation 11, we have

$$\|\hat{p}_n - p\|_2^2 \leq 4 \max_{f \in \mathcal{F}^*} \left| \mathbb{E}_p(f) - \frac{1}{n} \sum_{i=1}^n f(X_i) \right| + \|p_n^* - p\|_2^2. \tag{13}$$

We first analyze the term $\|p_n^* - p\|_2^2$ in equation 13. From Proposition A.1, we have that

$$p = \prod_{V' \in \mathcal{C}(\mathcal{G})} \psi_{V'} \circ e_{V'}$$

and there exists some $C_\psi > 0$ so that $\psi_{V'} \leq C_\psi$ for all $V'$ and that, for some $L_\psi$, all $\psi_{V'}$ are $L_\psi$-Lipschitz continuous. We pick a sufficiently large $C$ that is greater than $C_\psi$. Also, by the definition of $\mathcal{F}^*$ in equation 9, we can pick a $q_{V'} \in \mathcal{F}_{V'}$ for each $V' \in \mathcal{C}(\mathcal{G})$ and form an $f \in \mathcal{F}^*$ such that

$$f = \prod_{V' \in \mathcal{C}(\mathcal{G})} q_{V'} \circ e_{V'}.$$

We will specify each $q_{V'}$ later. Then, we have

$$\|f - p\|_\infty = \left\| \prod_{V' \in \mathcal{C}(\mathcal{G})} q_{V'} \circ e_{V'} - \prod_{V' \in \mathcal{C}(\mathcal{G})} \psi_{V'} \circ e_{V'} \right\|_\infty$$

$$\leq C^{|\mathcal{C}(\mathcal{G})|-1} \sum_{V' \in \mathcal{C}(\mathcal{G})} \left\| q_{V'} \circ e_{V'} - \psi_{V'} \circ e_{V'} \right\|_\infty \quad \text{by Lemma B.2} \qquad (14)$$

Recall that $\mathcal{F}_{V'} = \mathcal{F}(\ell_{V'}, w_{V'}, s, C)$. For any sufficiently large $m, N \in \mathbb{N}$ which we will determine later, we pick

$$\ell_{V'} = 8 + (m+5)(1 + \lceil \log_2 |V'| \rceil),$$
$$w_{V'} = (|V'|, 6(|V'|+1)N, 6(|V'|+1)N, \ldots, 6(|V'|+1)N, 1),$$
$$s = \lfloor 141(r+2)^{r+3} N(m+6) \rfloor$$

and recall that we have picked $C$ to be a constant larger than $C_\psi$ before. It is easy to check that the hypotheses of Theorem B.3 are satisfied with $K = L_\psi$, $\beta = 1$ and $d = |V'|$ and hence, by Theorem B.3, if we pick

$$q_{V'} = \arg \min_{q'_{V'} \in \mathcal{F}_{V'}} \left\| q'_{V'} \circ e_{V'} - \psi_{V'} \circ e_{V'} \right\|_\infty$$

then we have

$$\left\| q_{V'} \circ e_{V'} - \psi_{V'} \circ e_{V'} \right\|_\infty \leq (2L_\psi + 1)(1 + |V'|^2 + 1)6^{|V'|} N 2^{-m} + L_\psi 3 N^{-1/|V'|}$$

$$= O(N 2^{-m} + N^{-1/r}) \qquad (15)$$

By plugging equation 15 into equation 14, we have

$$\|f - p\|_\infty \leq C^{|\mathcal{C}(\mathcal{G})|-1} \sum_{V' \in \mathcal{C}(\mathcal{G})} O(N 2^{-m} + N^{-1/r}) = O(N 2^{-m} + N^{-1/r})$$

Recall that the domain is $[0,1]^d$ and hence we have

$$\|f - p\|_2^2 = \int_{[0,1]^d} |f(x) - p(x)|^2 dx \leq \|f - p\|_\infty^2 .$$

Now, by the optimality of $p_n^*$ in equation 10, we have

$$\|p_n^* - p\|_2^2 \leq \|f - p\|_2^2 \leq \|f - p\|_\infty^2 = O(N^2 2^{-2m} + N^{-2/r}). \qquad (16)$$

Now, we take care of the term $\max_{f \in \mathcal{F}^*} \left| \mathbb{E}_p(f) - \frac{1}{n} \sum_{i=1}^n f(X_i) \right|$ in equation 13. To bound this term for all $f \in \mathcal{F}^*$, we first construct an $\epsilon$-cover of $\mathcal{F}^*$ in $L^\infty$. Then, we use the Hoeffding's inequality to bound this term for each $f$ in the $\epsilon$-cover and use the union bound to control the total failure probability. To construct an $\epsilon$-cover, we define the following notations. For any $V' \in \mathcal{C}(\mathcal{G})$, let $\widetilde{\mathcal{F}}_{V'}$ be a minimal $\frac{\epsilon}{C^{|\mathcal{C}(\mathcal{G})|-1}}$-cover of $\mathcal{F}_{V'}$ in $L^\infty$ where $\epsilon$ is a sufficiently small value and we will determine it later. Also, let

$$\widetilde{\mathcal{F}}^* = \left\{ \prod_{V' \in \mathcal{C}(\mathcal{G})} \tilde{q}_{V'} \circ e_{V'} \mid \tilde{q}_{V'} \in \widetilde{\mathcal{F}}_{V'} \right\}. \qquad (17)$$

We will show that $\widetilde{\mathcal{F}}^*$ is an $\epsilon$-cover of $\mathcal{F}$ in $L^\infty$. For any $f \in \mathcal{F}$, it can be expressed as

$$f = \prod_{V' \in \mathcal{C}(\mathcal{G})} q_{V'} \circ e_{V'} \quad \text{for some } q_{V'} \in Q_{V'}$$

Since $\widetilde{\mathcal{F}}_{V'}$ is an $\frac{\epsilon}{C^{|\mathcal{C}(\mathcal{G})|-1}}$-cover of $\mathcal{F}_{V'}$ in $L^\infty$ for all $V' \in \mathcal{C}(\mathcal{G})$, there exists a $\tilde{q}_{V'} \in \widetilde{\mathcal{F}}_{V'}$ such that

$$\|q_{V'} - \tilde{q}_{V'}\|_\infty \leq \frac{\epsilon}{C^{|\mathcal{C}(\mathcal{G})|-1}}.$$

By the definition of $\widetilde{\mathcal{F}}$, we set $\tilde{f} \in \widetilde{\mathcal{F}}$ to be

$$\tilde{f} = \prod_{V' \in \mathcal{C}(\mathcal{G})} \tilde{q}_{V'} \circ e_{V'}$$

By Lemma B.2, we check that

$$\left\| f - \tilde{f} \right\|_\infty = \left\| \prod_{V' \in \mathcal{C}(\mathcal{G})} q_{V'} - \prod_{V' \in \mathcal{C}(\mathcal{G})} \tilde{q}_{V'} \right\|_\infty = C^{|\mathcal{C}(\mathcal{G})|-1} \cdot \sum_{V' \in \mathcal{C}(\mathcal{G})} \|q_{V'} - \tilde{q}_{V'}\|_\infty$$

$$\leq C^{|\mathcal{C}(\mathcal{G})|-1} \cdot \frac{\epsilon}{C^{|\mathcal{C}(\mathcal{G})|-1}}$$

$$= \epsilon.$$

Now, we return to the term $\max_{f \in \mathcal{F}} \left| \mathbb{E}_p(f) - \frac{1}{n} \sum_{i=1}^n f(X_i) \right|$. Since $\widetilde{\mathcal{F}}^*$ is an $\epsilon$-cover of $\mathcal{F}^*$ in $L^\infty$, for any $f \in \mathcal{F}^*$, there exists a $\tilde{f} \in \widetilde{\mathcal{F}}^*$ such that $\|f - \tilde{f}\|_\infty \leq \epsilon$ and we have

$$\left| \mathbb{E}_p(f) - \frac{1}{n} \sum_{i=1}^n f(X_i) \right|$$

$$\leq \left| \mathbb{E}_p(f) - \mathbb{E}_p(\tilde{f}) \right| + \left| \mathbb{E}_p(\tilde{f}) - \frac{1}{n} \sum_{i=1}^n \tilde{f}(X_i) \right| + \left| \frac{1}{n} \sum_{i=1}^n \tilde{f}(X_i) - \frac{1}{n} \sum_{i=1}^n f(X_i) \right|$$

$$\leq 2\epsilon + \left| \mathbb{E}_p(\tilde{f}) - \frac{1}{n} \sum_{i=1}^n \tilde{f}(X_i) \right|$$

which implies

$$\max_{f \in \mathcal{F}^*} \left| \mathbb{E}_p(f) - \frac{1}{n} \sum_{i=1}^n f(X_i) \right| \leq 2\epsilon + \max_{\tilde{f} \in \widetilde{\mathcal{F}}^*} \left| \mathbb{E}_p(\tilde{f}) - \frac{1}{n} \sum_{i=1}^n \tilde{f}(X_i) \right|. \tag{18}$$

By Hoeffding's inequality and the union bound, for any $t > 0$, the probability of

$$\max_{\tilde{f} \in \widetilde{\mathcal{F}}^*} \left| \mathbb{E}_p(\tilde{f}) - \frac{1}{n} \sum_{i=1}^n \tilde{f}(X_i) \right| > t$$

is bounded by $|\widetilde{\mathcal{F}}^*| \cdot e^{-\Omega(nt^2)}$.

To bound the term $|\widetilde{\mathcal{F}}^*|$, by the definition of $\widetilde{\mathcal{F}}^*$ in equation 17, we first have

$$\log |\widetilde{\mathcal{F}}^*| = \sum_{V' \in \mathcal{C}(\mathcal{G})} \log |\widetilde{\mathcal{F}}_{V'}|.$$

For each term $\log |\widetilde{\mathcal{F}}_{V'}|$, by Lemma B.4, we have

$$\log |\widetilde{\mathcal{F}}_{V'}| \leq (s+1) \log(2^{2L_{V'}+5} \epsilon^{-1}(L_{V'}+1)|V'|^2 s^{2L_{V'}}).$$

We now bound the architecture parameters. Recall that

$$\ell_{V'} = 8 + (m+5)(1 + \lceil \log_2 |V'| \rceil) \text{ for any } V' \in \mathcal{C}(\mathcal{G}) \text{ and}$$

$$s = \lfloor 141(r+2)^{r+3} N(m+6) \rfloor.$$

Namely, we have

$$\ell_{V'} = O(m) \quad \text{and} \quad s = O(Nm) \quad \text{which implies} \quad \log |\widetilde{\mathcal{F}}_{V'}| \leq O(Nm^2 \log \frac{Nm}{\epsilon}).$$

That means we have

$$\log |\widetilde{\mathcal{F}}^*| \leq \sum_{V' \in \mathcal{C}(\mathcal{G})} O(Nm^2 \log \frac{Nm}{\epsilon}) = O(Nm^2 \log \frac{Nm}{\epsilon}).$$

By setting $t = O(\sqrt{\frac{Nm^2}{n} \log \frac{Nnm}{\epsilon}})$, we have

$$\max_{\tilde{f} \in \widetilde{\mathcal{F}}^*} \left| \mathbb{E}_p(\tilde{f}) - \frac{1}{n} \sum_{i=1}^n \tilde{f}(X_i) \right| < O(\sqrt{\frac{Nm^2}{n} \log \frac{Nnm}{\epsilon}})$$

with at least probability $1 - |\widetilde{\mathcal{F}}^*| \cdot e^{-\Omega(nt^2)} \to 1$ as $n \to \infty$. Plugging it into equation 18, we have

$$\max_{f \in \mathcal{F}^*} \left| \mathbb{E}_p(f) - \frac{1}{n} \sum_{i=1}^n f(X_i) \right| \leq 2\epsilon + O(\sqrt{\frac{Nm^2}{n} \log \frac{Nnm}{\epsilon}}). \tag{19}$$

Furthermore, by plugging equation 16 and equation 19 into equation 13, we have

$$\|\hat{p}_n - p\|_2^2 \leq 4 \max_{f \in Q} \left| \mathbb{E}_p(f) - \frac{1}{n} \sum_{i=1}^n f(X_i) \right| + \|p_n^* - p\|_2^2$$

$$< O(\epsilon + \sqrt{\frac{Nm^2}{n} \log \frac{Nnm}{\epsilon}} + N^2 2^{-2m} + N^{-2/r}).$$

By picking

$$\epsilon = n^{-\frac{2}{r+4}}, \quad N = n^{\frac{r}{r+4}} \quad \text{and} \quad m = \frac{r+1}{r+4} \log n,$$

we have

$$\|\hat{p}_n - p\|_2^2 \leq \widetilde{O}(n^{-\frac{2}{r+4}}).$$

$\square$

## C PROOF OF THEOREM 4.8

**Theorem C.1.** *Let $\mathcal{G} = (V, E)$ be a finite graph. There exists an estimator $V_n$ such that for any positive Lipschitz continuous density $p$ satisfying the Markov property with respect to a graph $\mathcal{G}$, we have that*
$$\|p - V_n\|_1 \in \widetilde{O}_p\left(n^{-1/(2+r)}\right),$$
*where $V_n$ is a function of $n$ iid samples from $p$, and $r$ is the size of the largest clique in $\mathcal{G}$.*

**Lemma C.2.** *Let $(\Omega, \Sigma, \mu)$ be a measure space, and let $f_1, \ldots, f_m$ and $g_1, \ldots, g_m$ be measurable and absolutely integrable functions on $\Omega$. Further suppose there exists a constant $C \geq 0$ such that, for all $i \in [m]$,*
$$\|f_i\|_\infty \leq C \quad \text{and} \quad \|g_i\|_\infty \leq C.$$
*Then the following inequality holds:*
$$\left\| \prod_{i=1}^m f_i - \prod_{i=1}^m g_i \right\|_1 \leq C^{m-1} \sum_{i=1}^m \|f_i - g_i\|_1.$$

The proof of this lemma will rely heavily on the 1-$\infty$ form of Hölder's Inequality. The following is such a version of Hölder's Inequality, as stated in (Folland, 1999, Theorem 6.8a).

**Theorem C.3** (Hölder's Inequality). *If $f$ and $g$ are measurable functions on a measure space $(\Omega, \Sigma, \mu)$, then*

$$\|f \cdot g\|_1 \leq \|f\|_1 \|g\|_\infty \,.$$

*Proof of Lemma C.2.* We will proceed by induction on $m$.

**Case $m = 1$:** Trivial.

**Induction:** Suppose the lemma holds for some value of $m$. A consequence of Hölder's Inequality is that for general functions $f$ and $g$, with $\|f\|_1, \|f\|_\infty, \|g\|_1, \|g\|_\infty$ finite, both $\|f \cdot g\|_1$ and $\|f \cdot g\|_\infty$ are also finite. From the inductive hypothesis we have that

$$\left\| \prod_{i=1}^{m+1} f_i - \prod_{i=1}^{m+1} g_i \right\|_1 \leq \left\| \prod_{i=1}^{m} f_i \cdot f_{m+1} - \prod_{i=1}^{m} g_i \cdot f_{m+1} \right\|_1 + \left\| \prod_{i=1}^{m} g_i \cdot f_{m+1} - \prod_{i=1}^{m} g_i \cdot g_{m+1} \right\|_1$$

$$\leq \left\| \prod_{i=1}^{m} f_i - \prod_{i=1}^{m} g_i \right\|_1 \|f_{m+1}\|_\infty + \left\| \prod_{i=1}^{m} g_i \right\|_\infty \|f_{m+1} - g_{m+1}\|_1$$

$$\leq \left\| \prod_{i=1}^{m} f_i - \prod_{i=1}^{m} g_i \right\|_1 C + C^m \|f_{m+1} - g_{m+1}\|_1$$

$$\leq C^{m-1} \sum_{i=1}^{m} \|f_i - g_i\|_1 C + C^m \|f_{m+1} - g_{m+1}\|_1$$

$$\leq C^m \sum_{i=1}^{m+1} \|f_i - g_i\|_1 \,.$$

$\square$

**Lemma C.4.** *Let $p$ be an $L$-Lipschitz probability density on $[0,1]^d$ then $\|p\|_\infty \leq 1 + L\sqrt{d}$.*

*Proof of Lemma C.4.* Since $p$ is $L$-Lipschitz, we have

$$|p(x) - p(y)| \leq L \cdot \|x - y\|_2 \quad \text{for any } x, y \in [0,1]^d$$
$$\leq L\sqrt{d}.$$

Also, since $p$ is a probability density, we have

$$1 = \|p\|_1 = \int_{x \in [0,1]^d} p(x)dx \geq \min_{x \in [0,1]^d} p(x).$$

Combining these two inequalities, we have

$$\|p\|_\infty = \max_{x \in [0,1]^d} p(x) \leq \min_{x \in [0,1]^d} p(x) + L\sqrt{d} \leq 1 + L\sqrt{d}.$$

$\square$

**Lemma C.5.** *Let $1 \geq \epsilon > 0$ and $C \geq 1$. Then,*

$$N(\mathbb{Q}_{d,b,V,C}, \epsilon) \leq (2C/\epsilon)^{\left(b^{|V|}\right)} \,.$$

*Proof of Lemma C.5.* Given an $\epsilon \in (0,1]$, consider the set

$$\widetilde{\mathbb{Q}}_{d,b,V,C,\epsilon} := \left\{ x \mapsto \sum_{A \in [b]^{|V|}} w_A \mathbf{1}_{\Lambda_{d,b,A,V}}(x) \mid w_A \in \{0, \epsilon, 2\epsilon, \ldots, \lfloor C/\epsilon \rfloor \epsilon\} \right\}.$$

Clearly, $\widetilde{\mathbb{Q}}_{d,b,V,C,\epsilon} \subset \mathbb{Q}_{d,b,V,C}$. We have that $|\{0, \epsilon, 2\epsilon, \ldots, \lfloor C/\epsilon \rfloor \epsilon\}| \leq 1 + C/\epsilon \leq 2C/\epsilon$. Thus, from a simple combinatorics argument, it follows that

$$\left| \widetilde{\mathbb{Q}}_{d,b,V,C,\epsilon} \right| \leq (2C/\epsilon)^{\left(b^{|V|}\right)} \,.$$

Now, we argue that $\widetilde{\mathbb{Q}}_{d,b,V,C,\epsilon}$ is an $\epsilon$-cover of $\mathbb{Q}_{d,b,V,C}$ in $L^1$ distance. Let $q \in \mathbb{Q}_{d,b,V,C}$. From the definition of $\mathbb{Q}_{d,b,V,C}$, it follows that

$$q(x) = \sum_{A \in [b]^{|V|}} w_A \mathbf{1}_{\Lambda_{d,b,A,V}}(x) \quad \text{for all } x \in [0,1]^d.$$

From the definition of $\widetilde{\mathbb{Q}}_{d,b,V,C,\epsilon}$, there exists a $\tilde{q} \in \widetilde{\mathbb{Q}}_{d,b,V,C,\epsilon}$, with

$$\tilde{q}(x) = \sum_{A \in [b]^{|V|}} \tilde{w}_A \mathbf{1}_{\Lambda_{d,b,A,V}}(x) \quad \text{for all } x \in [0,1]^d,$$

where $|w_A - \tilde{w}_A| \le \epsilon$ for all $A$. It therefore follows that

$$\|q - \tilde{q}\|_1 = \int_{[0,1]^d} \left| \sum_{A \in [b]^{|V|}} w_A \mathbf{1}_{\Lambda_{d,b,A,V}}(x) - \sum_{A \in [b]^{|V|}} \tilde{w}_A \mathbf{1}_{\Lambda_{d,b,A,V}}(x) \right| dx$$

$$= \int_{[0,1]^d} \left| \sum_{A \in [b]^{|V|}} (w_A - \tilde{w}_A) \mathbf{1}_{\Lambda_{d,b,A,V}}(x) \right| dx$$

$$= \sum_{A \in [b]^{|V|}} |w_A - \tilde{w}_A| \int_{[0,1]^d} \mathbf{1}_{\Lambda_{d,b,A,V}}(x) dx$$

$$\le \sum_{A \in [b]^{|V|}} \epsilon \cdot \frac{1}{b^{|V|}}$$

$$= \epsilon.$$

$\square$

**Lemma C.6.** *Let $V \subset [d]$ and $f : [0,1]^{|V|} \mapsto \mathbb{R}$ be an L-Lipschitz function with $0 \le f \le C$ for some $C$. Then,*

$$\min_{q \in \mathbb{Q}_{d,b,V,C}} \|f \circ e_V - q\|_1 \le \sqrt{|V|} L/(2b).$$

*Proof of Lemma C.6.* For any $A \in [b]^{|V|}$, recall that $\Lambda_{|V|,b,A,[|V|]}$ is defined in equation 3 as

$$\Lambda_{|V|,b,A,[|V|]} = \prod_{i=1}^{|V|} \left[ \frac{A_i - 1}{b}, \frac{A_i}{b} \right]$$

and let $\lambda_A$ be the centroid of $\Lambda_{|V|,b,A,[|V|]}$ Also, let $q$ be the function on $[0,1]^d$ such that

$$q(x) = \sum_{A \in [b]^{|V|}} f(\lambda_A) \mathbf{1}_{\Lambda_{d,b,A,V}}(x) \quad \text{for all } x \in [0,1]^d.$$

By the assumption of $f \le C$, we have $q \in \mathbb{Q}_{d,b,V,C}$. Now, we bound $\|f \circ e_V - q\|_1$.

$$\|f \circ e_V - q\|_1 = \int_{x \in [0,1]^d} \left| f(e_v(x)) - \sum_{A \in [b]^{|V|}} f(\lambda_A) \mathbf{1}_{\Lambda_{d,b,A,V}}(x) \right| dx$$

$$= \int_{x \in [0,1]^{|V|}} \left| f(x) - \sum_{A \in [b]^{|V|}} f(\lambda_A) \mathbf{1}_{\Lambda_{|V|,b,A,[|V|]}}(x) \right| dx \quad \text{by Tonelli's Theorem}$$

$$\le \sum_{A \in [b]^{|V|}} \int_{x \in \Lambda_{|V|,b,A,[|V|]}} |f(x) - f(\lambda_A) \mathbf{1}_{\Lambda_{|V|,b,A,[|V|]}}(x)| dx. \tag{20}$$

Note that if $x \in \Lambda_{|V|,b,A,[|V|]}$ then $\mathbf{1}_{\Lambda_{|V|,b,A,[|V|]}}(x) = 1$. Hence, for $x \in \Lambda_{|V|,b,A,[|V|]}$, we have

$$|f(x) - f(\lambda_A) \mathbf{1}_{\Lambda_{|V|,b,A,[|V|]}}(x)| = |f(x) - f(\lambda_A)| \le L\|x - \lambda_A\|_2 \le L \cdot \frac{\sqrt{|V|}}{2b}.$$

Plugging this into equation 20, we have

$$\|f \circ e_V - q\|_1 \leq \sum_{A \in [b]^{|V|}} \int_{x \in \Lambda_{|V|,b,A,[|V|]}} L \cdot \frac{\sqrt{|V|}}{2b} dx = L \cdot \frac{\sqrt{|V|}}{2b}.$$

$\square$

**Theorem C.7** (Theorem 3.4 page 7 of Ashtiani et al. (2018), Theorem 3.6 page 54 of Devroye & Lugosi (2001))**.** *There exists a deterministic algorithm that, given a collection $\mathcal{C}$ of distributions $\{p_1, \ldots, p_M\}$, a parameter $\varepsilon > 0$ and at least $\frac{\log\left(3M^2/\delta\right)}{2\varepsilon^2}$ iid samples from an unknown distribution $p$, outputs an index $j \in [M]$ such that*

$$\|p_j - p\|_1 \leq 3 \min_{i \in [M]} \|p_i - p\|_1 + 4\varepsilon \tag{21}$$

*with probability at least $1 - \delta/3$.*

*Proof of Theorem C.1.* For this proof, we will be employing Theorem C.7. For a nonnegative integrable function $f$ on $[0,1]^d$, define $\bar{N} : f \mapsto f/\|f\|_1$, with $\bar{N}(0)$ being set to the constant uniform density.

For any $d, b \in \mathbb{N}$, $V' \subset [d]$ and sufficiently large constant $C > 0$, recall that $\mathbb{Q}_{d,b,V',C}$ is the set of histograms of width $b$ on $V'$ whose maximum weight is at most $C$ as defined in equation 4 and let $\widetilde{\mathbb{Q}}_{d,b,V',C,b^{-1}}$ be a minimal $b^{-1}$ cover of $\mathbb{Q}_{d,b,V',C}$. Also, recall that $\mathcal{C}(\mathcal{G})$ is the set of maximal cliques in $\mathcal{G}$. Let

$$Q_n = \left\{ \prod_{V' \in \mathcal{C}(\mathcal{G})} q_{V'} \mid q_{V'} \in \mathbb{Q}_{d,b,V',C} \right\} \quad \text{and} \quad \widetilde{Q}_n = \left\{ \prod_{V' \in \mathcal{C}(\mathcal{G})} \tilde{q}_{V'} \mid \tilde{q}_{V'} \in \widetilde{\mathbb{Q}}_{d,b,V',C,b^{-1}} \right\}. \tag{22}$$

The collection $\mathcal{C}$ of densities from Theorem C.7 correspond to the set $\bar{N}(\widetilde{Q}_n) := \{\bar{N}(q) \mid q \in \widetilde{Q}_n\}$.

To show that Theorem C.7 applies, we will first show that, for sufficiently large $n$,

$$n \geq \frac{\log\left(3M^2/\delta\right)}{2\varepsilon^2}.$$

We first give a bound on $M$. Note that

$$M = \left|\bar{N}\left(\widetilde{Q}_n\right)\right| \leq \left|\widetilde{Q}_n\right| = \prod_{V' \in \mathcal{C}(\mathcal{G})} \left|\widetilde{\mathbb{Q}}_{d,b,V',C}\right|.$$

Since each $\widetilde{\mathbb{Q}}_{d,b,V',C}$ is a minimal $b^{-1}$ cover of $\mathbb{Q}_{d,b,V',C}$, we have

$$\left|\widetilde{\mathbb{Q}}_{d,b,V',C}\right| = N(\mathbb{Q}_{d,b,V',C}, b^{-1}) \quad \text{by the definition of a } b^{-1}\text{-cover.}$$

By Lemma C.5 and $|V'| \leq r$, we have

$$N(\mathbb{Q}_{d,b,V',C}, b^{-1}) \leq (2Cb)^{\left(b^{|V'|}\right)} \leq (2Cb)^{b^r}.$$

It implies that

$$M \leq \prod_{V' \in \mathcal{C}(\mathcal{G})} (2Cb)^{b^r} \leq (2Cb)^{|\mathcal{C}(\mathcal{G})| \cdot b^r} \leq (2Cb)^{2^d \cdot b^r} \quad \text{since } |\mathcal{C}(\mathcal{G})| \leq 2^{|V|} = 2^d.$$

By applying a logarithm yields, we have

$$\log M \leq 2^d b^r \log(2Cb) = O(b^r \log b)$$

Now, we have

$$\frac{\log\left(3M^2/\delta\right)}{2\varepsilon^2} = \frac{1}{2\epsilon^2} \cdot (\log 3 + 2\log M + \log(1/\delta)) \leq O(\frac{1}{\epsilon^2}(b^r \log b + \log \frac{1}{\delta})).$$

By picking

$$\epsilon = n^{-\frac{1}{r+2}}\log n, \quad b = n^{-\frac{1}{r+2}} \quad \text{and} \quad \delta = \frac{1}{n},\tag{23}$$

we have, for a sufficiently large $n$, $\frac{\log(3M^2/\delta)}{2\varepsilon^2} \leq n$.

Now, we are going to examine the RHS of equation 21 in Theorem C.7, i.e. bound the term $\min_{\tilde{q}\in\bar{N}(\widetilde{Q}_n)}\|p - \tilde{q}\|_1$. We first bound the term $\min_{\tilde{q}\in\widetilde{Q}_n}\|p - \tilde{q}\|_1$ and return to $\bar{N}(\widetilde{Q}_n)$ later. Note that

$$\min_{\tilde{q}\in\widetilde{Q}_n}\|p - \tilde{q}\|_1 \leq \min_{q\in Q_n}\left(\|p - q\|_1 + \min_{\tilde{q}\in\widetilde{Q}_n}\|q - \tilde{q}\|_1\right).$$

Recall the definition of $Q_n$ and $\widetilde{Q}_n$ in equation 22. We have, for each $q \in Q_n$, there is a $\tilde{q} \in \widetilde{Q}_n$ such that, by Lemma C.2,

$$\|\tilde{q} - q\|_1 = O\left(\frac{C^{|\mathcal{C}(\mathcal{G})|-1}}{b}\right) = \widetilde{O}(n^{-\frac{1}{2+r}}).\tag{24}$$

Therefore, we have

$$\min_{\tilde{q}\in\widetilde{Q}_n}\|p - \tilde{q}\|_1 \leq \min_{q\in Q_n}\|p - q\|_1 + \widetilde{O}(n^{-\frac{1}{2+r}}).$$

Now, we investigate the term $\min_{q\in Q_n}\|p - q\|_1$. From Proposition A.1 it follows that

$$p(x) = \prod_{V'\in\mathcal{C}(\mathcal{G})} \psi_{V'} \circ e_{V'} \quad \text{where all } \psi_{V'} \text{ are all Lipschitz continuous for some } L.\tag{25}$$

Because $\psi_{V'}$ are all Lipschitz continuous on a bounded set, they must all be bounded and, for sufficiently large $n$, $\psi_{V'} \leq C$ for all $V' \in \mathcal{C}(\mathcal{G})$. By equation 25 and the definition in equation 22, we can express

$$\min_{q\in Q_n}\|p - q\|_1 = \min_{q\in Q_n}\left\|\prod_{V'\in\mathcal{C}(\mathcal{G})} \psi_{V'} \circ e_{V'} - q\right\|_1 = \min_{q_{V'}\in\mathbb{Q}_{d,b,V',C}}\left\|\prod_{V'\in\mathcal{C}(\mathcal{G})} (\psi_{V'} \circ e_{V'} - q_{V'})\right\|_1\tag{26}$$

By Lemma C.2, for any $q_{V'} \in \mathbb{Q}_{d,b,V',C}$, we have

$$\left\|\prod_{V'\in\mathcal{C}(\mathcal{G})} (\psi_{V'} \circ e_{V'} - q_{V'})\right\|_1 \leq C^{d-1} \sum_{V'\in\mathcal{C}(\mathcal{G})} \left\|\psi_{V'} \circ e_{V'} - q_{V'}\right\|_1$$

and, by Lemma C.6, we have

$$\min_{q_{V'}\in\mathbb{Q}_{d,b,V',C}} \left\|\psi_{V'} \circ e_{V'} - q_{V'}\right\|_1 \leq \frac{L}{b}.$$

Plugging them into equation 26, we have

$$\min_{q\in Q_n}\|p - q\|_1 \leq \sum_{V'\in\mathcal{C}(\mathcal{G})} C^{d-1} \cdot \frac{L}{b} = \widetilde{O}(n^{-\frac{1}{r+2}}).$$

If $q^*$ is minimizer of $\arg\min_{\tilde{q}\in Q_n}\|p - \tilde{q}\|_1$, it also implies that

$$|\|q^*\|_1 - 1| \leq \|p - q^*\|_1 = \widetilde{O}(n^{-\frac{1}{r+2}})\tag{27}$$

Combining with equation 24, if $\tilde{q}^*$ is a minimizer of $\arg\min_{q\in\widetilde{Q}_n}\|p - q\|_1$, we have

$$|\|\tilde{q}^*\|_1 - 1| = \widetilde{O}(n^{-\frac{1}{r+2}}) \quad \text{which means } \|\tilde{q}^*\|_1 \to 1 \text{ as } n \to \infty.$$

Note that $\widetilde{Q}_n$ may contain the 0 function. The fact that $\|\tilde{q}^*\|_1 \to 1$ suggests that, for a sufficiently large $n$, the 0 function is not a minimizer of $\arg\min_{\tilde{q}\in\widetilde{Q}_n}\|p - \tilde{q}\|_1$. For any $n \in \mathbb{N}$, let $\widetilde{Q}'_n$ the set $\widetilde{Q}_n$ with 0 removed, i.e.,

$$\widetilde{Q}'_n = \begin{cases} \widetilde{Q}_n & \text{if } 0 \notin \widetilde{Q}_n \\ \widetilde{Q}_n \setminus \{0\} & \text{if } 0 \in \widetilde{Q}_n. \end{cases}$$

Hence, for a sufficiently large $n$, we have $\min_{\tilde{q} \in \widetilde{Q}'_n} \|p - \tilde{q}\|_1 = \min_{\tilde{q} \in \widetilde{Q}_n} \|p - \tilde{q}\|_1$. Now, we are ready to analyze the term $\min_{\tilde{q} \in \bar{N}(\widetilde{Q}_n)} \|p - \tilde{q}\|_1$. We have

$$\min_{\tilde{q} \in \bar{N}(\widetilde{Q}_n)} \|p - \tilde{q}\|_1 \leq \min_{\tilde{q} \in \bar{N}(\widetilde{Q}'_n)} \|p - \tilde{q}\|_1 \leq \min_{\tilde{q} \in \widetilde{Q}'_n} \|p - \bar{N}(\tilde{q})\|_1 \leq \min_{\tilde{q} \in \widetilde{Q}'_n} \left( \|p - \tilde{q}\|_1 + \|\tilde{q} - \bar{N}(\tilde{q})\|_1 \right).$$

For any $\tilde{q} \in \widetilde{Q}'_n$, we have

$$\left\|\tilde{q} - \bar{N}(\tilde{q})\right\|_1 = \left\|\tilde{q} - \frac{\tilde{q}}{\|\tilde{q}\|_1}\right\|_1 = |1 - \|\tilde{q}\|_1| = |\|p\|_1 - \|\tilde{q}\|_1| \leq \|p - \tilde{q}\|_1.$$

Hence, by $\min_{\tilde{q} \in \widetilde{Q}'_n} \|p - \tilde{q}\|_1 = \min_{\tilde{q} \in \widetilde{Q}_n} \|p - \tilde{q}\|_1$ and equation 27, we have

$$\min_{\tilde{q} \in \bar{N}(\widetilde{Q}_n)} \|p - \tilde{q}\|_1 \leq 2 \min_{\tilde{q} \in \widetilde{Q}'_n} \|p - \tilde{q}\|_1 = \widetilde{O}(n^{-\frac{1}{r+2}}).$$

Recall that we set $\varepsilon = n^{-\frac{1}{r+2}} \log(n)$ in equation 23. Finally, suppose $q'$ is the output of the algorithm, we have

$$\|q' - p\|_1 \leq 3 \cdot \min_{\tilde{q} \in \bar{N}(\widetilde{Q}_n)} \|p - \tilde{q}\|_1 + 4 \cdot \epsilon = \widetilde{O}(n^{-\frac{1}{r+2}}).$$

$\square$

# D   GRAPH PROOFS

For any $d, d', t \in \mathbb{N}$, define $L_{d \times d'}$ to be the graph whose vertex set is $[d] \times [d']$ and edge set is

$$\{((i,j), (i',j')) \mid i, i' \in [d], j, j' \in [d'], (i,j) \neq (i',j'), |i - j| + |i' - j'| \leq 1\}$$

and $L_{d \times d'}^t$ to be the graph whose vertex set is $[d] \times [d']$ and edge set is

$$\{((i,j), (i',j')) \mid i, i' \in [d], j, j' \in [d'], (i,j) \neq (i',j'), |i - j| + |i' - j'| \leq t\}.$$

For any $d, d', t \in \mathbb{N}$, define $L_{d \times d'}^+$ to be the graph whose vertex set is $[d] \times [d']$ and edge set is

$$\{((i,j), (i',j')) \mid i, i' \in [d], j, j' \in [d'], (i,j) \neq (i',j'), \max\{|i - j|, |i' - j'|\} \leq 1\}$$

and $(L_{d \times d'}^+)^t$ to be the graph whose vertex set is $[d] \times [d']$ and edge set is

$$\{((i,j), (i',j')) \mid i, i' \in [d], j, j' \in [d'], (i,j) \neq (i',j'), \max\{|i - j|, |i' - j'|\} \leq t\}.$$

For any $d, t \in \mathbb{N}$, define $L_d$ to be the graph whose vertex set is $[d]$ and edge set is $\{(i,j) \mid i \neq j, |i - j| \geq 1\}$ and $L_d^t$ to be the graph whose vertex set is $[d]$ and edge set is $\{(i,j) \mid i \neq j, |i - j| \geq t\}$.

*Proof of Lemma 4.3.* For any clique $C$ in $(L_{d \times d'})^t$, let $(i_0, j_0)$ (resp. $(i_1, j_1)$, $(i'_0, j'_0)$ and $(i'_1, j'_1)$) be the vertex in $C$ such that $i_0 + j_0$ is maximal (resp. $i_1 + j_1$ is minimal, $i'_0 - j'_0$ is maximal and $i'_1 - j'_1$ is minimal). Namely, the vertex set of $C$ is a subset of

$$S := \{(i,j) \mid i \in [d], j \in [d'], i_1 + j_1 \leq i + j \leq i_0 + j_0, i'_1 - j'_1 \leq i - j \leq i'_0 - j'_0\}.$$

By the definition of cliques and $(L_{d \times d'})^t$, we have

$(i_0 + j_0) - (i_1 + j_1) \leq |i_0 - i_1| + |j_0 - j_1| \leq t$   since there is an edge between $(i_0, j_0)$ and $(i_1, j_1)$
$(i'_0 - j'_0) - (i'_1 - j'_1) \leq |i'_0 - i'_1| + |j'_0 - j'_1| \leq t$   since there is an edge between $(i'_0, j'_0)$ and $(i'_1, j'_1)$

To bound the size of $S$, we observe that, for each of the at most $t + 1$ possible values $i_1 + j_1, i_1 + j_1 + 1, \ldots, i_0 + j_0$ equal to $i + j$, there are at most $\lceil \frac{t+1}{2} \rceil$ possible values among $i'_1 + j'_1, i'_1 + j'_1 + 1, \ldots, i'_0 + j'_0$ equal to $i - j$ by considering the parity. Therefore, $|S|$ is at most $(t + 1) \cdot \lceil \frac{t+1}{2} \rceil$.

Hence, the size of the largest clique in $(L_{d \times d'})^t$ is at most $(t + 1) \cdot \lceil \frac{t+1}{2} \rceil \leq \frac{t^2 + 4t + 3}{2}$.   $\square$

*Proof of Lemma 4.4.* It is easy to check that the subgraph of $(L_{d \times d}^+)^t$ induced by the vertex set $[t+1] \times [t+1]$ is a clique. Hence, the size of the largest clique in $(L_{d \times d'}^+)^t$ is at least $(t+1)^2$.

For any clique $C$ in $(L_{d \times d'}^+)^t$, let $i_0$ (resp. $i_0'$) be the smallest (resp. largest) first index of the vertices in $C$ and $j_0$ (resp. $j_0'$) be the smallest (resp. largest) second index of the vertices in $C$. Namely, the vertex set of $C$ is a subset of

$$S := \{(i,j) | i \in [d], j \in [d'], i_0 \le i \le i_0', j_0 \le j \le j_0'\}.$$

To bound the size of $S$, by the definition of cliques and $(L_{d \times d'}^+)^t$, we have

$$i_0' - i_0 \le t \quad \text{and} \quad j_0' - j_0 \le t$$

Therefore, $|S|$ is at most $(t+1)^2$.

Hence, the size of the largest clique in $(L_{d \times d'}^+)^t$ is $(t+1)^2$.

$\square$

*Proof of Lemma 4.6.* It is easy to check that the subgraph of $L_d^t$ induced by the vertex set $[\min\{t+1, d\}]$ is a clique. Hence, the size of the largest clique in $L_d^t$ is at least $\min\{t+1, d\}$.

For any clique $C$ in $L_d^t$, let $i_0$ (resp. $j_0$) be the smallest (resp. largest) index of the vertex in $C$. Namely, the vertex set of $C$ is be a subset of $S := \{i | i \in [d], i_0 \le i \le j_0\}$. By the definition of cliques and $L_d^t$, we have $|i - j| \le \min\{t, d-1\}$. Therefore, $|S|$ is at most $\min\{t+1, d\}$.

Hence, the size of the largest clique in $L_d^t$ is $\min\{t+1, d\}$.

$\square$

# E    LOWER BOUND FOR MRF RATES

We approach this problem assuming the data domain is $[0,1]^d$. For any MRF graph $\mathcal{G}$ with maximum clique size $r$, no estimator can achieve a rate of $O\left(n^{-1/(2+r-\varepsilon)}\right)$ for the set of all Lipschitz continuous densities, for any $\varepsilon > 0$. We prove this by contradiction.

Suppose there exists a graph $\mathcal{G}$, $\varepsilon > 0$, and an estimator $\hat{p}$ that achieves this rate on Lipschitz continuous densities satisfying the Markov property with respect to $\mathcal{G}$. Without loss of generality, assume the first $r$ entries of the random vector, $X_1, \ldots, X_r$, form a maximal clique in $\mathcal{G}$.

Consider an arbitrary $r$-dimensional Lipschitz continuous density $q$ and let $q'$ be the density where, for $Y \sim q'$, $(Y_1, \ldots, Y_r) \sim q$ and $Y_{r+1}, \ldots, Y_d$ are i.i.d. uniform random variables on $[0,1]$, jointly independent of $(Y_1, \ldots, Y_r)$. Note that $q'$ is a Lipschitz continuous density satisfying the Markov property with respect to $\mathcal{G}$.

Using $\hat{p}$ to estimate $q'$, we get $\|q' - \hat{p}\|_1 \in O\left(n^{-1/(2+r-\varepsilon)}\right)$. Let $\mathcal{L}$ denote the law of a random variable, e.g., $\mathcal{L}(Y) = q'$.

It is well-known that applying the same function to a pair of random variables never increases their $L^1$ distance (see Devroye & Lugosi (2001), Section 5.4). Let $f : (x_1, \ldots, x_d) \mapsto (x_1, \ldots, x_r)$. Let $\hat{X} \sim \hat{p}$. We then have $\mathcal{L}(f(Y)) = q$ and:

$$
\begin{aligned}
\left\| q - \mathcal{L}(f(\hat{X})) \right\|_1 &= \left\| \mathcal{L}(f(Y)) - \mathcal{L}(f(\hat{X})) \right\|_1 \\
&\le \left\| \mathcal{L}(Y) - \mathcal{L}(\hat{X}) \right\|_1 \\
&= \|q' - \hat{p}\|_1 \\
&= O\left(n^{-1/(2+r-\varepsilon)}\right)
\end{aligned}
$$

Thus, $\mathcal{L}(f(\hat{X}))$ is an estimator that achieves a rate of $O\left(n^{-1/(2+r-\varepsilon)}\right)$ on $r$-dimensional Lipschitz continuous densities. However, it is known that no estimator can achieve this rate, leading to a contradiction.

## F    COCO SCATTER PLOTS

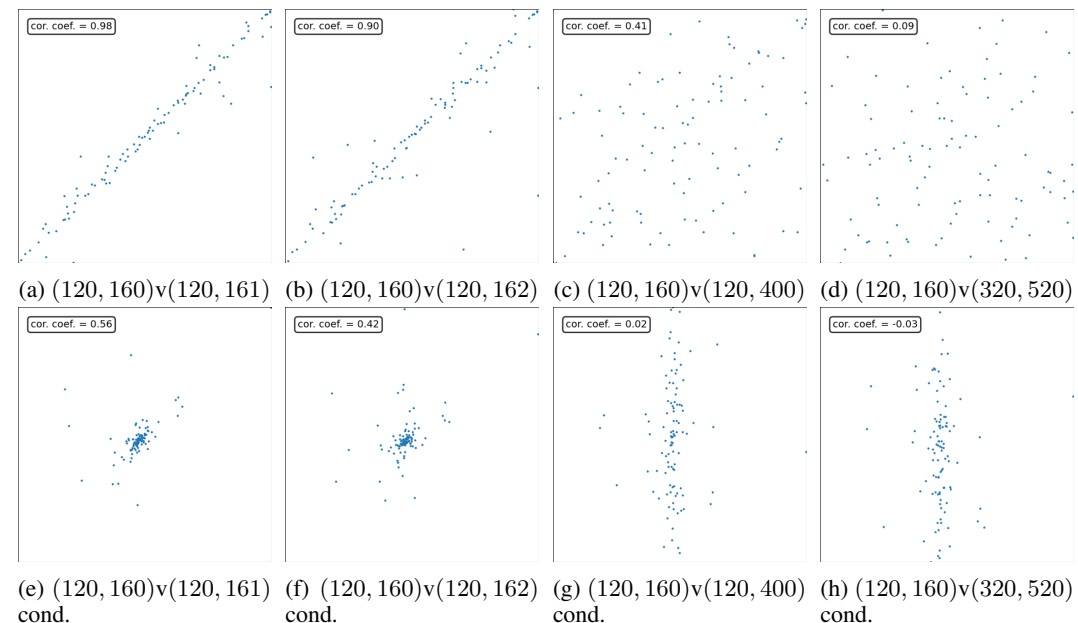

(a) $(120, 160)$v$(120, 161)$  (b) $(120, 160)$v$(120, 162)$  (c) $(120, 160)$v$(120, 400)$  (d) $(120, 160)$v$(320, 520)$

(e) $(120, 160)$v$(120, 161)$ cond.  (f) $(120, 160)$v$(120, 162)$ cond.  (g) $(120, 160)$v$(120, 400)$ cond.  (h) $(120, 160)$v$(320, 520)$ cond.

Figure 8: This figure presents scatter plots analogous to those in Figure 3 of the main text, but derived from the COCO training set (Lin et al., 2014). The conditional scatter plots are based on pixel (121,160) being near its median value.

Due to memory constraints, we used a subset of the data:

1. 4000 random samples were initially selected.
2. From these, 100 images with pixel (121,160) nearest to the median were chosen for the conditional plots.

Note that increasing the sample size for conditioning resulted in lower observed correlation. This is because a larger sample allows for a more precise conditioning, better approximating the true conditional distribution. The wider the range of values for the conditioning pixel (121,160), the more the selected points resemble the unconditional distribution, potentially introducing spurious correlation.

These experiments provide an approximation of the conditional data. In our observations, using larger datasets consistently and significantly reduced the observed correlation. This suggests that using an even larger dataset would likely further reduce the observed correlation, bringing the results closer to the true conditional independence structure.

