# OpenReview forum: "Dimension-Independent Rates for Structured Neural Density Estimation"
_ICLR.cc/2025/Conference — Submitted to ICLR 2025_

### Official Review · Reviewer_Xtr6 · 2024-11-02

**Soundness:** 2
**Presentation:** 3
**Contribution:** 4
**Rating:** 8
**Confidence:** 3

**Summary:**

The authors study the upper bounds of the sample requirements for probability density estimation using deep learning. They establish an upper bound on the $L_1$ error for probability density estimation based on the size of the largest clique in the undirected graph of data, i.e., the Markov random field (MRF). Furthermore, for one-dimensional or two-dimensional array data, the authors present an upper bound on the $L_1$ error, deriving an upper bound on the size of cliques of the data. They also provide proofs for these theoretical results.

**Strengths:**

* To the best of my knowledge, the upper bound on the estimation error for density estimation based on the structure of the undirected graph model is novel. The theoretical results of this study are considered important for understanding the sample requirements of density estimation for high-dimensional data.
* Proofs are provided for these theoretical results, and no major errors have been found.

**Weaknesses:**

#### Major weaknesses:
* The empirical evidence is limited for supporting the author's conclusions regarding the MRF structure of image data. Thus, it is still unclear whether the sample requirements in Corollary 4.5 align with those for image data based solely on the experimental results of this study.
* Additionally, it is desirable to conduct numerical experiments using synthetic data that supports the upper bounds of $L_1$ error for probability density estimation based on the size of the largest clique.

#### Minor weaknesses:
* Line 016-017: The statement “this size is typically independent of the data dimensionality” is more appropriate than “this size is typically constant, i.e., $r = O(1)$” in the lines, as this study does not focus on cases where the data dimensionality approaches infinity.
* Lines 419-422: A more detailed explanation or definition of $L_{d \times d'}$ and $L^+_{d \times d'}$ would clarify their specific meanings.

**Questions:**

* You suggest that the image data is a MRF such that $L_{d \times d'}$ and $L_{d \times d'}^+$, $L_{d \times d'}^2$, or
$(L_{d \times d'}^+)^2$ (lines 258-290 and lines 444-451).
     According to Corollary 4.5, this implies that the sample requirement for density estimation of the image data is at most $O(n^{-1/9})$.
     However, it seems slightly challenging to reliably reach this conclusion solely based on the results presented in Figure 3. Could you provide references to any research that has empirically tested this claim using image data, thereby supporting the upper bound presented in the corollary? Alternatively, are there any experimental methodologies that could confirm this upper bound?
* In lines 1402-1403, which theoretical results or resarch does "However, it is known that no estimator can achieve this rate" refer to?
* The discription in lines 446-451 is confusing. Does it mean that, assuming that CIFAR-10 is a MFR of $(L_{32 \times 32}^+)^2$, the probability density of the images can be estimated with the data sample requirement of $(L_{32 \times 32}^+)^2$, i.e., $n^{-1/9}$ in Corollary 4.5?

---

> ### Author Response · Authors · 2024-11-21
> **Response to Reviewer Xtr6**
>
> __”The empirical evidence is limited for supporting the author's conclusions regarding the MRF structure of image data. Thus, it is still unclear whether the sample requirements in Corollary 4.5 align with those for image data based solely on the experimental results of this study.”__
>
> __”...it seems slightly challenging to reliably reach this conclusion solely based on the results presented in Figure 3. Could you provide references to any research that has empirically tested this claim using image data, thereby supporting the upper bound presented in the corollary? Alternatively, are there any experimental methodologies that could confirm this upper bound?__
>
> __”Additionally, it is desirable to conduct numerical experiments using synthetic data that supports the upper bounds of error for probability density estimation based on the size of the largest clique.”__
>
> We agree that our evidence does not definitively justify the MRF model. In fact, natural image data likely doesn't perfectly satisfy the MRF assumption. Nonetheless, our evidence strongly suggests that something close to the MRF model holds, and there is extensive precedent for modeling images as MRFs in the image processing literature—please see the "General Author Response". We believe the theoretical conclusions of this work are most novel and striking in this context. Regarding additional experiments, please see the "General Author Response".
>
> __”In lines 1402-1403, which theoretical results or resarch does ‘However, it is known that no estimator can achieve this rate’ refer to?”__
>
> This refers to the classical nonparametric rate for estimating Lipschitz continuous densities. These rates are well-known for the $L^2$ risk (e.g. [1-2] below); for the case of the $L^1$ risk corresponding results can be found in [3-4].
>
> [1] Charles J Stone. Optimal rates of convergence for nonparametric estimators. The Annals of Statistics, 8(6):1348–1360, 1980.
>
> [2] A.B. Tsybakov. Introduction to nonparametric estimation. Springer Series in Statistics, New York, 2009.
>
> [3] Theorem 1, L. Devroye and L. Gyorfi. Nonparametric Density Estimation: The L1 View. Wiley Interscience Series in Discrete Mathematics. Wiley, 1985.
>
> [4] Theorem 6.3.8, Evarist Giné and Richard Nickl. Mathematical foundations of infinite-dimensional statistical models. Number 40. Cambridge University Press, 2016.
>
> __”The discription in lines 446-451 is confusing. Does it mean that, assuming that CIFAR-10 is a MFR of $(L_{32\times 32}^+)^2$, the probability density of the images can be estimated with the data sample requirement of  $(L^+_{32\times 32})^2$, i.e., $n^{-1/9}$ in Corollary 4.5?“__
>
> You are correct. We will re-word this to be more clear.
>
> We will be sure to incorporate your suggestions in "minor weaknesses".

---

### Official Review · Reviewer_eiDw · 2024-11-02

**Soundness:** 3
**Presentation:** 3
**Contribution:** 2
**Rating:** 6
**Confidence:** 4

**Summary:**

The author(s) analyze a general class of densities that are markov to an undirected graph. Under the sense of density estimation, they derive the convergence rate of neural networks. The rate only depends on the effective dimension of the graph, i.e. the maximium clique size r.

**Strengths:**

It's a joy to read the paper. The paper proposed a sound theorem on dimension-free convergence rate, which looks right to me. The paper clearly claimed the assumption for the convergence rate theorem, and justified the assumption and its connection to the real world applications. They put a lot of discussion on the assumption and explain the gap between the assumption.

**Weaknesses:**

Though a lot of discussion on the assumption, the paper lacks the numerical results (example or real world application) to illustrate the convergence rate. As mentioned in line 511, it lacks the proof when the optimal rate happens.

**Questions:**

I'm confused about the figure 3. The author(s) try to claim that the dependency of pixels are reduced given the observations of a neighboring pixel. Can author(s) explain how MRF is applied in this scenario and also provide experiment on the convergence rate with neural networks (better with MRF), in order to justify how MRF helps bypass the curse of dimensionality?

---

> ### Author Response · Authors · 2024-11-21
> **Response to Reviewer eiDw**
>
> __”Though a lot of discussion on the assumption, the paper lacks the numerical results (example or real world application) to illustrate the convergence rate. As mentioned in line 511, it lacks the proof when the optimal rate happens.”__
>
> Please see the “General Author Response” regarding experiments.
>
> __”I'm confused about the figure 3. The author(s) try to claim that the dependency of pixels are reduced given the observations of a neighboring pixel. Can author(s) explain how MRF is applied in this scenario and also provide experiment on the convergence rate with neural networks (better with MRF), in order to justify how MRF helps bypass the curse of dimensionality?”__
>
> Figure 3 supports our argument that a grid MRF $(L_{w,h})^k$ reasonably models CIFAR images by demonstrating two key consequences of the MRF assumption:
> * As pixels become more spatially distant, their dependence weakens (confirmed in the top row of Figure 3).
> * Two pixels $a$ and $b$ should be independent when conditioned on pixels surrounding $a$ (depending on $k$). The intuition is that $a$ should contain no more information about $b$ than the collection of pixels surrounding $a$. Since testing this directly is challenging, we made two simplifications:
>     1. We condition on just one adjacent pixel $a'$ rather than all surrounding pixels, testing if this neighboring pixel captures most of the information in $a$.
>     2. We fix $a'$ to a specific value rather than testing independence across all values
>
> Even with simplification 1, we observe that conditioning strongly reduces the dependence between $a$ and $b$, supporting our claim of strong local dependence and weak distant dependence.
>
> We do not claim this definitively confirms the MRF model—indeed, it is unlikely to be perfectly satisfied. However, our findings support MRFs as a useful framework for understanding image statistics. Please see our "General Author Response" for more on the MRF assumption, particularly its widespread acceptance in the image processing community.

---

### Official Review · Reviewer_3xmu · 2024-11-02

**Soundness:** 3
**Presentation:** 3
**Contribution:** 3
**Rating:** 8
**Confidence:** 3

**Summary:**

The paper studies classical density estimation under the assumption that the data has markov random field structure. They show that there exist betwork architectures such that the curse of dimensionality (type $n^{-c/d}$ convergence rates, for $c$ some constant) is overcome by an ERM procedure if the dependency graph structure is sufficiently benign. I.e., if the largest clique is $O(r)$, the effective dimension is $r$ not $d$.

**Strengths:**

* The authors offer an interesting pespective on ambient dimensionality via MRF clique size.

* Overall I found the paper well-written/structured and easy to follow

*I believe the results relating clique size to convergence rate are novel

* The paper is well-contextualized (i.e., wrt manifold hypothesis) and written with ample motivation with examples in mind.

* The power graph construction is interesting

**Weaknesses:**

* I am not entirely convinced about dependence (in terms of graph hops, mixing or other related concepts) accurately capturing the hardness of a problem as opposed to support on low-dimensional manifold. See my question Q3 below.

*Comment:  While I could infer the assumptions of the main theorem from the text preceding it, the assumptions are stated too implicitly for my taste. I think in particular the "markov property with respect to G" should be explicitly defined somewhere in the text (ideally with \cref in thm 4.2 to ease readibility).

**Questions:**

Q1. Is it assumed that p is realizable by the hypothesis class in question in thm 4.2?

Q2. Could you comment on your section modeling CIFAR10 as an MRF. It is not obvious to me from your discussion following Corollary 4.5 that either hop structure is appropriate for modeling the ground truth distribution. Are there known results characterizing the locality of dependence on CIFAR (genuine question --- have very little background on image classification)?

Q3. I am not sure whether the MRF/clique assumption is so different from the manifold hypothesis, and would be interested to hear further comments on this. The example I have in mind is the following (which has both low clique size and low effective dimension --- neither of which actually capture the convergence rate of ERM which actually decreases with $d$). Consider, the Gaussian  autoregression $X_{t+1} = aX_t +W_t$ (say for a fix sequence length $d$) does not just have global dependence between all coordinates $X_t$ but is actually easier to learn for $|a| \geq 1$ (more dependency/no mixing --- see the classical result from Mann&Wald 1943 and note that regression is equivalent to density estimation in KL for this model).  Note also that if $|a| >> 1$ the process is concentrated on the last coordinates, making dimensionality relevant instead. Overall, this kind of begs the question to me what the actual notion of signal-noise actually is in density estimation?

---

> ### Author Response · Authors · 2024-11-21
> **Response to Reviewer 3xmu**
>
> __”I am not entirely convinced about dependence... accurately capturing the hardness of a problem as opposed to support on low-dimensional manifold”__
>
> __”The example I have in mind is the following (which has both low clique size and low effective dimension -- neither of which actually capture the convergence rate of ERM which actually decreases with $d$)...”__
>
> Thanks for the intriguing reference and example; it will be useful to keep in mind going forward.
>
> To begin, we emphasize that the MRF assumption provides a view of high-dimensional data that is, in some sense, _orthogonal_ to the manifold hypothesis. __Keeping in mind that MRF assumption corresponds to the Markov graph not being complete__ (some edges missing), and we can demonstrate all four possible combinations of MRF/manifold hypothesis satisfaction in 2D, where the MRF assumption reduces to X and Y being independent:
> * MRF true, MH true: $X \sim \text{unif}[0,1]$, $Y \sim \text{unif}[0,0.01]$,$X$ and $Y$ independent (your example: density concentrated in coordinate-aligned subspace)
> * MRF true, MH false: $X,Y \sim \text{unif}[0,1]$, $X$ and $Y$ independent (density not concentrated)
> * MRF false, MH true: $X \sim \text{unif}[0,1]$, $X=Y$
> * MRF false, MH false: $X,Y \sim N(0,1)$, weakly correlated
>
> This generalizes to higher dimensions with more complex graphs, where the manifold hypothesis satisfaction becomes a matter of intrinsic dimension rather than binary.
> Regarding images, our experiments (Figure 3, and Appendix F for COCO) show that distant pixels are weakly correlated, with correlation essentially vanishing when conditioned on nearby pixels (noting that higher resolution images require greater pixel separation). While some theoretical work exists on using conditional independence to improve nonparametric density estimation [1], theoretically justifying this for MRFs in the context of images is novel.
>
> Please also see our “General Author Response.”
>
> __”Is it assumed that p is realizable by the hypothesis class in question in thm 4.2?”__
>
> No; p need only satisfy Lipschitz continuity, positivity, the MRF assumption, and have compact support. From a learning theory perspective, the hypothesis class grows with $n$ (increasing width, depth, etc.). For any fixed $n$, the hypothesis space consists of a single neural network architecture over all possible parameter values. As $n$ increases, the network grows, yielding a richer hypothesis space. More formally, this is a sieve estimator approximating the class of Lipschitz densities.
>
> __”Could you comment on your section modeling CIFAR10 as an MRF. It is not obvious to me from your discussion following Corollary 4.5 that either hop structure is appropriate for modeling the ground truth distribution. Are there known results characterizing the locality of dependence on CIFAR...”__
>
> It would be helpful if you could elaborate a bit. We can see many possible ways to interpret your question, including:
> * Is the general concept of modeling natural images as an MRF reasonable?
> * Is CIFAR-10 _specifically_ is reasonably modeled as an MRF?
> * Is the specific MRF in Cor 4.5 appropriate (maybe a larger exponent)?
> * Are local patches of images _actually_ highly dependent.
>
> We provide a general response that should address your concerns. Please also see our "General Author Response" regarding the extensive foundation for MRF-based image modeling in the image processing literature.
>
> There is substantial work characterizing the structure of $m\times m$ image patches in natural images. For instance, "Emergence of simple-cell receptive field properties by learning a sparse code for natural images" (Nature 1996) found these regions have "sparse representations" in appropriate bases. As we noted:
>
> >Further supporting this hypothesis, Carlsson et al. (2008) discovered that the set of $3\times × 3$ pixel patches from natural images concentrates around a 2-dimensional manifold.
>
> These patches typically correspond to oriented edges or textures—features that neural networks learn to detect (see Figure 3 the NeurIPS 2012 AlexNet paper). The key insight is the limited degrees of freedom: within an image patch, a small collection of pixels largely determines the remainder, indicating strong dependence.
> While this geometric perspective is well-studied, we're unaware of probabilistic or information-theoretic approaches testing this. Our Figure 3 demonstrates pixel decorrelation with distance (especially when conditioned on nearby pixels). While we know of no prior work explicitly investigating this phenomenon, it would likely be unsurprising to the image processing community. We can provide correlation heatmaps if helpful.
> Regarding CIFAR-10's specific dependence structure or rigorous justification for the exponent 2 in Corollary 4.5, we know of no targeted studies—most work addresses natural images broadly.
>
> [1] Nonparametric estimation of component distributions in a multivariate mixture. Hall and Zhou, 2003

---

> > ### Comment · Reviewer_3xmu · 2024-11-21
> >
> > Many thanks for the detailed response to my question and an interesting follow-up. I see no reason not to accept this paper and will raise my score accordingly.

---

### Official Review · Reviewer_Zqnm · 2024-11-05

**Soundness:** 3
**Presentation:** 3
**Contribution:** 2
**Rating:** 5
**Confidence:** 3

**Summary:**

This paper studies learning distributions (ie generative models) under the assumption that the Markov Random Field (MRF) generating the data has no large cliques. [A MRF has an associated graph $G$ which captures the conditional independence stucture of the data $x$: if any path from $i$ to $j$ in G goes through $k$, then $x_i$ and $x_j$ are conditionally indepndent given $k$. Crucially, the distribution p(x) can be written as a product of terms that depend only on $x_S$, where S is a subset of nodes in a clique of G.] The main result (Theorem 4.2) shows that the distribution can be learned in $n$ samples up to TV distance $n^{-1/(4+r)}$ where $r$ is the size of the largest clique in $G$ by ERM over a class of NNs.

This paper provides an alternative viewpoint to the manifold hypothesis for why generative models can perform well without needing a number of samples exponential in the ambient dimension. The authors posit that the MRF assumption may better capture the stucture in the data than the manifold hypothesis when many parts of the data are [conditionally] independent: for example pixels far apart in images, or words far apart in language.

**Strengths:**

- The paper is well-written and very clear, including the proofs.
- The motivation of wanting to consider structural assumptions that go beyond the manifold hypothesis is sound.
- While the MRF assumption is certainly idealistic, the authors suggest in their conclusion that ultimately these 2 could be viewed in conjunction, since they capture orthogonal types of structure in the data.

**Weaknesses:**

My main concern is the theoretical contibutions are quite weak (and this is a theory paper)/
- The main result shows that Lipshitz MRFs can be learned by ERM over $\prod_{S \in clique(G)} \psi_S(x_S)$ neural networks, where $x_S$ is $x$ restricted to the inputs in the set $S$, and each $\psi_S$ is a neural network, achieving a rate of $n^{-1/(r + 4)}$. However, it is known how to learn such Lipshitz MRFs at a better rate of $n^{-1/(r + 2)}$, with a different (non NN-based) algorithm. While this algorithm may be computationally intractable, ERM over NNs is not necessarily tractable.
- The proof of the main result (Thrm 4.2) seem to follow closely from results in Schmidt-Hieber 2017 which gives approximation of Liphsitz functions by NNs. I am unaware if there is any significant technical novelty in using this for MRFs.
- The consequences in section 4.3 are all quite trivial.

Discussion of Related work lacking:
- There is no discussion of the literature on learning MRFs. I am not an expert in the area but this is glaringly lacking, and I do wonder if the authors are reinventing the wheel with Theorem 4.8.
- Can the MRF be learned if the graph $G$ is unknown? The authors should discuss the literature here.


In light of some of the discussions with the authors, I am willing to see this paper accepted, though I feel the theoretical result should be better contextualized, including discussion of the following:

(1) Other work on non-parametric density estimation, in particular which avoids the curse of dimensionality. As in the authors' comments, they should explain which are subsumed by the MRF or Manifold hypothesis.
(2) The difference between this work and the well-studied with MRFs, which typically involves learning the graph structure (though typically with additional parameteric assumptions).
(3) Why do the authors get a $n^{-1/(r + 4)}$ rate for neural networks? Can NN possibly achieve the $n^{-1/(r + 2)}$ rate, or are they intrinsically limited? Other works (eg. https://arxiv.org/abs/2212.13848) achieve the minimax $n^{-1/(d + 2)}$ rate for learning Lipschitz functions on $R^d$.

Ultimately, I still feel the theoretical contribution is non-surprising and not-particularly challenging: the paper shows that under the MRF assumption with clique size $r$ --- meaning the density can be written as a product of Lipshitz functions on $r$ coordinates --- density estimation can be done at the $n^{-1/(r + 2)}$ rate, matching the rate for Lipshitz density estimation in $r$ dimensions. If the density function is fit with neural networks, the rate achieved is slightly worse: $n^{-1/(r + 4)}$, while other works (eg. see above) in non-parametric learning with neural-networks do achieve the minimax rate.

**Questions:**

See above.

---

> ### Author Response · Authors · 2024-11-21
> **Response to Reviewer Zqnm**
>
> __”My main concern is the theoretical contibutions are quite weak”__
>
> __”The proof of the main result (Thrm 4.2) seem to follow closely from results in Schmidt-Hieber 2017 which gives approximation of Liphsitz functions by NNs. I am unaware if there is any significant technical novelty in using this for MRFs.”__
>
> The main technical novelty lies in (a) avoiding Lipschitz assumptions on the potentials, (b) a nontrivial analysis of the covering numbers for families of MRF densities that delivers dimension-independent rates. Regarding (a), while the analysis would be quite trivial if one simply assumes that the clique potentials in $p = \prod_{V'} \psi_{V'}$ are all Lipschitz continuous, proceeding with only the assumption that $p$ is Lipschitz is more challenging. (Note that in practice the clique potentials are unknown and untestable, so it would be artificial to impose assumptions on them.)
>
> We believe that this analysis, combined with the practical connection of MRFs to high-dimensional data such as images, is novel and significant to the deep learning (and hence ICLR) community—a point with which the other reviewers seem to agree.
>
> __”There is no discussion of the literature on learning MRFs. I am not an expert in the area but this is glaringly lacking, and I do wonder if the authors are reinventing the wheel with Theorem 4.8.”__
>
> __”Can the MRF be learned if the graph $G is unknown? The authors should discuss the literature here”.__
>
> The comments regarding nonparametric MRF learning are well-taken. While we are not aware of any results on density estimation as in Theorem 4.8, there is relevant literature we should discuss, including results on structure learning in special cases such as nonparanormal graphical models and general non-Gaussian models. We will incorporate this discussion in a revision.
>
> Notably, none of these results cover density estimation and more importantly do not establish dimension-independent rates. Another notable feature of our results is the use of neural networks, which is more relevant in contemporary applications.
>
> ### Nonparanormal
> Regularized rank-based estimation of high-dimensional nonparanormal graphical models. Xue and Zou, 2013
>
>
> High Dimensional Semiparametric Gaussian Copula Graphical Models. Liu et al., 2012
>
>
> Sparse Nonparametric Graphical Models. Lafferty et al., 2013
>
>
> The Nonparanormal: Semiparametric Estimation of High Dimensional Undirected Graphs. Liu et al., 2009
>
> ### Non-Gaussian
> High-dimensional covariance estimation by minimizing $\ell_1$-penalized log-determinant divergence. Ravikumar et al., 2008
>
>
> Learning non-Gaussian graphical models via Hessian scores and triangular transport. Baptista et al., 2023
>
>
> Generalized Precision Matrix for Scalable Estimation of Nonparametric Markov Networks. Zheng et al., 2023

---

> > ### Comment · Reviewer_Zqnm · 2024-11-25
> > **Repsonse to Author comments**
> >
> > I looked though most of these references regarding the literature, and it seems these works are concerned with **learning the graph structure, which is the main reason why non-parametric MRF learning is hard.** If the graph structure is known, then the problem is much easier, and should reduce to standard non-parameteric density estimation. The authors should better investigate the literature to contexturalize it within the literature of non-parameteric density estimation. Its well known that NNs can represent Lipschitz functions, so "using neural networks" to do something we already know they can do is not a novel contribution in my opinion.
> >
> > In authors response, they highlight two points potential technical contributions (a) estimating a Lipschitz p with non-Lipschitz factors and (b) dimension independent covering numbers.
> > (b) seems completely standard given the graph structure, so I don't know what technical barriers were overcome, and
> > (a) In the proofs, its seems like this challenge of the $\psi_V$ not being Lipschitz is immediately mediated by a technical lemma Prop A.1 which shows that p being lipschitz immediately implies that the $\psi_V$ are Lipschitz.
> >
> > Overall, I still feel the technical contribution of this paper is **extremely** lacking especially given the **absence of proper contexturalization** within the related work. I am also reducing my score because on further inspection of the proofs, they are not clearly written [for example Proposition A.1: I could not find equation (2) or why it is true from the Chang lecture notes reference, and I am confused because $V'' \subset V'$, and yet $V'' \setminus V'$ is non-empty?].

---

> > > ### Author Response · Authors · 2024-11-29
> > > **Second Response to Reviewer Zqnm**
> > >
> > > > ”...learning the graph structure, _which is the main reason why non-parametric MRF learning is hard._ If the graph structure is known, then the problem is much easier, and should reduce to standard non-parameteric density estimation. The authors should better investigate the literature to contexturalize it within the literature of non-parameteric density estimation. Its well known that NNs can represent Lipschitz functions, so "using neural networks" to do something we already know they can do is not a novel contribution in my opinion.”
> > >
> > > Can the reviewer please clarify their definition of “non-parametric MRF learning”? This can mean two different things: 1) Learning the distribution (e.g. in TV distance), or 2) Learning the graph structure (e.g. in Hamming distance). Since we already have an extensive discussion of 1) in the paper (see Sec 2, esp. L92-105, L129-137), our response interpreted the reviewer’s comment as asking for a comparison with 2), which we acknowledge is missing.
> > >
> > > Thus, **it would be especially helpful if there are specific results the reviewer has in mind**. For example, there are results for learning parametric models such as Ising models (in TV distance), but this is quite far from our nonparametric setting. For 1), there is also a difference between proper and improper learning.
> > >
> > > To the best of our knowledge, the minimax rate of nonparametric density estimation given a known MRF is an open problem that has not been resolved, and moreover our particular contributions (which go above and beyond simply the minimax rate) have not appeared previously. We chose to concentrate on contrasting our results with the manifold hypothesis and its implications for deep learning, but of course there are other ways to compare and contrast our results and we are happy to expand our discussion if the reviewer can provide details.
> > >
> > > > I still feel the technical contribution of this paper is extremely lacking
> > >
> > > We respectfully disagree that our contribution should be evaluated purely on technical difficulty. Our main contribution is demonstrating that MRFs provide a compelling framework for explaining why neural networks succeed in learning high-dimensional distributions. Our results provide useful and rigorous insights for the deep learning community and ICLR more broadly.
> > >
> > > Indeed, this has been positively noted by other reviewers - for instance, Reviewer 3xmu called it "well-contextualized... with ample motivation with examples in mind" and noted we "offer an interesting perspective on ambient dimensionality via MRF clique size." Reviewer eiDw similarly praised how we "justified the assumption and its connection to real world applications."
> > >
> > > Finally, although our results assume the graph is known, this is because **in practice the graph is known**: We are  analyzing models with a _known_ graph that is widely accepted in the community for its ability to naturally model real-world data types (see General Author Response). We also propose practical modifications to improve existing models and compare our framework to other explanations for deep learning's success in high dimensions.
> > >
> > > > ”A.1: I could not find equation (2) or why it is true from the Chang lecture notes reference, and I am confused because $V’’\subset V’$, and yet $V’’\setminus V’$  is non-empty?]”
> > >
> > > Regarding the set difference notation - of course, since $V''\subset V'$, we meant $|V'\setminus V''|$. Thank you for this careful reading, this was a typographical error.
> > >
> > >
> > > Equation (2) from Chang is something of a folklore result from graphical models. The proof is straightforward via induction on $d$, noting $\psi = V$ to bridge the difference in notation. (2) clearly holds for $d=1$ so and by induction (2) holds up to $d-1$. We have $V_A = \frac{p_{A}}{\prod_{B\subsetneq A} V_B}$ and $ Q := \prod_{B\subsetneq A} V_B = \prod_{i=0}^{d-1} \prod_{B\subset A: |B| = i} V_B$. Using (2) on $V_B$, a short calculation shows that for all $C \subsetneq A$ with $|C| = r$ the exponent of $p_C$ in $\prod_{B\subset A: |B| = s} V_B$ is ${d-r \choose s-r}( -1)^{(s-r)}$. Including all the factors in $Q$, the binomial theorem tells us that $p_C$ has an exponent of $\sum_{s=r}^{d-1} {d-r\choose s-r}  (-1)^{(s-r)} =\sum_{i=0}^{d-r-1} {d-r\choose i}  (-1)^{i} = (-1)^{d-r+1}$.
> > >
> > > For example, a similar formulation can also be found in the lecture notes http://www.stat.yale.edu/~pollard/Courses/251.spring04/Handouts/Hammersley-Clifford.pdf, where the definition of $\Psi_A$ under item <4> combined with equation <6> gives (2). If you like we could use a weaker form of (2) that follows more directly from Chang: $V_A = \prod_{B \subset A } p_A(x)^k$ for real values $k$.

---

> > > > ### Comment · Reviewer_Zqnm · 2024-12-03
> > > >
> > > > Thank you for the technical clarifications on Prop. A.1.
> > > >
> > > > I unfortunately do not know any results on learning non-parametric MRFs in TV-distance when the graph structure is known. It seems that this formulation is not typically studied, which is why I suggest contextulizing the question of learning non-parametric MRFs in TV-distance with other similar results on non-paramteric learning of distributions in TV distance. Of course, the completely non-parametric d-dimensional case has been mentioned in the related work by the authors, but perhaps there are other structured non-parameteric settings that have been studied?

---

> > > > > ### Author Response · Authors · 2024-12-03
> > > > > **Follow-up to Reviewer Zqnm**
> > > > >
> > > > > Thanks for your comments and clarification. Without specific suggestions from the reviewer, it is difficult to go into any detail without exhaustively covering the literature. For example, other structural assumptions include monotonicity, convexity, log-concave, sparsity, mixtures, and additive models. Crucially, with the exception of additive models and sparsity, **these assumptions do not address the curse of dimensionality.**
> > > > >
> > > > > We chose to focus on the manifold hypothesis since it is one of the most common structural assumptions used to break the curse of dimensionality, and arguably the most widely used explanation for why deep neural networks perform well on high-dimensional data. Additivity is a very strong assumption, much stronger than our MRF assumption, and sparsity is just a special case of the manifold hypothesis (e.g. the manifold is a linear subspace or union of linear subspaces).
> > > > >
> > > > > We would be happy to include some representative citations for each of these in the camera ready. For example, at L90, before Sec 2.1, we could add the following:
> > > > > > "In our discussion, we choose to focus on the manifold hypothesis since it is one of the most common structural assumptions used to break the curse of dimensionality and to explain the success of deep learning. However, it is worth pointing out other structural assumptions that have been studied such as monotonicity [1], convexity [2], log-concave [3], sparsity [4], mixtures [5], and additive models [6]."
> > > > >
> > > > >
> > > > > [1] P. Groeneboom. Estimating a monotone density, 1985
> > > > >
> > > > > [2] Groeneboom, P., Jongbloed, G. and Wellner (2001). Estimation of a convex function: Characterizations and asymptotic theory
> > > > >
> > > > > [3] Recent Progress in Log-Concave Density Estimation, Richard J. Samworth 2018
> > > > >
> > > > > [4] Han Liu, John Lafferty, and Larry Wasserman. Sparse nonparametric density estimation in high dimensions using the rodeo. 2007.
> > > > >
> > > > > [5] Genovese, Christopher R., and Larry Wasserman. "Rates of convergence for the Gaussian mixture sieve." The Annals of Statistics 28.4 (2000): 1105-1127.
> > > > >
> > > > > [6] Additive Regression and Other Nonparametric Models, Charles J. Stone 1985

---

### Author Response · Authors · 2024-11-21
**General Author Response**

# General Author Response

Many thanks to our reviewers for their thoughtful reviews.

A significant challenge in writing this paper was to present it in a way that bridges theory with applications in deep learning and computer vision, while making both theoretical and applied perspectives accessible and convincing to the other side. We were pleased to see from the positive reviews that we were largely successful in this regard, as evidenced by comments such as:

* Zqnm: “The paper is well-written and very clear, including the proofs.”
* Zqnm: “The motivation of wanting to consider structural assumptions that go beyond the manifold hypothesis is sound.”
* 3xmu: “The authors offer an interesting pespective on ambient dimensionality via MRF clique size.”
* 3xmu: “Overall I found the paper well-written/structured and easy to follow”
* 3xmu: “The paper is well-contextualized (i.e., wrt manifold hypothesis) and written with ample motivation with examples in mind.”
* eiDw: “It's a joy to read the paper.”
* Xtr6: “The theoretical results of this study are considered important for understanding the sample requirements of density estimation for high-dimensional data.”



Overall, only two concerns were raised by multiple reviewers: the lack of experiments and the validity of using a Markov random field (MRF) to model images. We address other issues in individual reviewer rebuttals.
## Lack of Experiments

__eiDw: ”Though a lot of discussion on the assumption, the paper lacks the numerical results (example or real world application) to illustrate the convergence rate.”__

__Xtr6: “Additionally, it is desirable to conduct numerical experiments using synthetic data that supports the upper bounds of error for probability density estimation based on the size of the largest clique.”__

We appreciate the reviewers' concern about the limited empirical validation. While we acknowledge this limitation, we note that:

* The paper's primary contribution is theoretical: Proving rigorously that neural networks can achieve dimension-independent rates in density estimation under commonly accepted assumptions. This provides yet another compelling justification for using DNNs in practice.

* More generally, our results establish fundamental bounds on learning distributions with MRF structure. In this sense our contributions are twofold: 1) We establish a general statistical framework for learning high-dimensional distributions with dimension-independent rates, and 2) We provide evidence (i.e. proofs) and intuition that neural networks are part of this framework and also achieve dimension-independent rates.

* The local dependency structure we identify is already implicitly leveraged in successful practical methods, particularly in patch-based approaches to anomaly detection such as SoftPatch (NeurIPS 2022). These existing empirical successes provide indirect validation of our theoretical framework.


* Due to strict space constraints (we are already at the page limit), we focused on developing the theoretical and intuitive foundations thoroughly. A comprehensive empirical study would require significant additional space to properly evaluate different architectures, datasets, and parameter settings.



## Validity of the MRF model

__3xmu: “I am not entirely convinced about dependence (in terms of graph hops, mixing or other related concepts) accurately capturing the hardness of a problem as opposed to support on low-dimensional manifold...”__

__Xtr6L: “The empirical evidence is limited for supporting the author's conclusions regarding the MRF structure of image data.”__

This concern was posed quite differently by the reviewers, so a detailed response will be included in the individual responses. We acknowledge that our evidence does not definitively confirm the MRF model—indeed, it is unlikely to be perfectly satisfied. However, many behaviors one would expect from the MRF model do hold, supporting it as a useful and novel framework for understanding image statistics. Moreover, there is extensive precedent for modeling images as MRFs in the image processing literature—this approach is so well-established that there are entire textbooks [1,2,3] ([1] has over 3,000 citations) and numerous highly-cited papers on the subject (e.g., [4] with 500+ citations and [5] with 900+ citations), further validating the reasonableness of our MRF-based approach to image modeling.

[1] Markov random field modeling in image analysis. Li 2009

[2] Markov Random Fields for Vision and Image Processing. Blake et al. 2011

[3] Markov Random Fields in Image Segmentation. Kato 2012

[4} Markov Random Field Image Models and Their Applications to Computer Vision. German and Graffigne 1986

[5] Combining markov random fields and convolutional neural networks for image synthesis. Li and Wand 2016

---

### Meta-Review · Area_Chair_XkVJ · 2024-12-20

**Metareview:**

The authors studied learning distributions where the data is generated by a Markov Random Field (MRF) with a clique assumption. The condition is considered as an alternative to the manifold hypothesis, where a significantly improved rate of density estimation can be achieved.

Although the average score appears high, after reading into the discussions of reviewer Zqnm, and starting a further brief discussion with reviewers leading to a willingness to reduce their score by 3 points, this paper is far closer to borderline than the scores currently indicates.

One of the main criticism is on the tightness of the rates of -1/(r+4), which compares against the known minimax rates of a complete MRF of -1/(r+2). There are additional concerns regarding the novelty of the techniques beyond existing work, the complexity of learning the MRF, and a lack of comparison versus existing rates.

Although none of these issues are critical on its own, but if we put these all together, there are quite a bit of doubt on whether the contributions are sufficient. I believe there is still room for this paper to be improved, and addressing some of these concerns can push this paper easily over the borderline. Therefore at this point, I will recommend reject.

**Additional Comments On Reviewer Discussion:**

The discussion of reviewer Zqnm was the most helpful towards the decision, as it raised more doubts in my mind, and led me to initiate an additional discussion with reviewers. This led to a willingness to reduce scores after reviewing the results and discussion more carefully, and also confirm some of my views. As mentioned above, some of the concerns on the rate, novelty, and comparison with existing work were reiterated, albeit not completely agreed upon, but it was sufficient to lead to the decision.

---

### Decision · Program_Chairs · 2025-01-22

Reject